# Non-random mating patterns within and across education and mental and somatic health

**Fartein Ask Torvik** [1,2,3] ✉, **Hans Fredrik Sunde** [1], **Rosa Cheesman** [2], **Nikolai Haahjem Eftedal** [2], **Matthew C. Keller**[4], **Eivind Ystrom** [2,3] & **Espen Moen Eilertsen** [2]

Partners resemble each other in health and education, but studies usually examine one trait at a time in established couples. Using data from all Norwegian first-time parents ($N = 187,926$) between 2016–2020, we analyse grade point average at age 16, educational attainment, and medical records of 10 mental and 10 somatic health conditions measured 10 to 5 years before childbirth. We find stronger partner similarity in mental (median $r = 0.14$) than in somatic health conditions (median $r = 0.04$), with ubiquitous cross-trait correlations in mental health (median $r = 0.13$). High grade point average or education is associated with better partner mental (median $r = -0.16$) and somatic (median $r = -0.08$) health. Elevated mental health correlations (median $r = 0.25$) in established couples indicate convergence. Analyses of siblings and in-laws suggest that health similarity is influenced by indirect assortment based on related traits. Adjusting for grade point average or education reduces partner health correlations by 30–40%. These findings have implications for the distribution of risk factors among children, genetic studies, and intergenerational transmission.

Assortative mating, the non-random matching of partners, is commonly studied from the perspective of social inequalities. Strong assortment for educational attainment (EA) is well-documented across disciplines[1], and partners often resemble each other in mental and somatic health conditions[2,3]. Recently, there has been a revived interest in matching across traits[4]. This is important because people do not choose their partners based on one phenotype at a time but holistically. We build upon the well-established links between educational attainment and health[5,6] and investigate assortative mating patterns within and between these interconnected phenomena in population-wide data. This provides insight into the clustering of education and health within families.

A comprehensive catalogue of partner correlations, based on the UK Biobank, presented partner similarity in 133 phenotypes, including EA and symptoms of mental disorders[1]. However, previous studies are with few exceptions[2] limited to cohort samples with healthy volunteer selection bias[7]. Issues related to selective non-participation are amplified in studies of couples, as both partners need to participate. In addition, partner correlations are usually assessed at arbitrary relationship stages and may therefore reflect convergence in addition to initial matching, leaving it unclear to what degree partners are similar in mental health at the time of couple formation. Another line of research investigates correlations between partners' genetic risk for mental disorders. Assortment based on heritable mental disorders should lead to genetic correlations between partners, and since the genes are determined before the couples are formed, correlations should be independent of convergence. These studies report null findings for mental disorders[8,9], except for schizophrenia[10]. Such

[1]Centre for Fertility and Health, Norwegian Institute of Public Health, Oslo, Norway. [2]PROMENTA Research Center, Department of Psychology, University of Oslo, Oslo, Norway. [3]PsychGen Centre for Genetic Epidemiology and Mental Health, Norwegian Institute of Public Health, Oslo, Norway. [4]Institute for Behavioral Genetics, University of Colorado Boulder, Boulder, CO, USA. ✉e-mail: fartein.ask.torvik@fhi.no

findings could imply that mental health does not influence partner selection. Despite a century of research on assortative mating, it is still questioned whether there is "really assortative mating on the liability to psychiatric disorders"[11]. Even less is known about assortment for somatic health. Good somatic health is a desired trait in partners[12], but it is unclear how similar partners are in somatic compared to mental health. Our first aim was therefore to assess partner similarity in education and mental and somatic health using population-wide prospective data.

Cross-trait assortment refers to non-random matching across different traits in the two partners[13]. Due to the competition for mates and attractiveness trade-offs, one should expect partner correlations to arise across different generally attractive traits, such as income and body mass[14]. The econometric research on cross-trait assortment has centred on such trade-offs[14,15], whereas the genetic research has seen cross-trait assortment as a source of genetic correlations[4,16]. Assortment across traits can lead to correlations between genetic[4] and environmental influences on different traits[17], which in turn can contribute to comorbidity and familial clustering of multiple disorders[18]. Beyond genuine increases in correlations between genetic liabilities, cross-trait assortment can also violate assumptions and bias genome-wide association and Mendelian randomization studies[19]. However, we are not aware of any studies examining cross-trait assortment for education and health phenotypes in representative samples. Addressing this gap, our second aim was to determine the degree of cross-trait assortment for education and a broad selection of health conditions.

Partner similarity can arise from several potentially co-occurring processes. In Fig. 1, we outline these processes and the role they play in the present paper. First, direct assortment (or primary phenotypic assortment) means that partners resemble each other in a trait because the observed trait influences partner selection (panel A). Direct assortment is a sufficient explanation for partner similarity in height[20–22]. Second, indirect assortment (also called secondary assortment) refers to similarity in a trait resulting from selection on a correlated trait, which may be unknown (panel B) or known (panel C). For instance, similarity in a specific mental disorder could arise from assortment on psychiatric vulnerabilities. If one trait, such as attractiveness, underlies assortment for multiple other traits, cross-trait assortment can be observed for these other traits. Direct assortment on an imperfectly measured phenotype can statistically resemble indirect assortment on an unobserved phenotype[20]. In such cases, assortment may be said to be direct for the trait of interest, but indirect for the indicator. Third, social stratification (or social homogamy) refers to individuals selecting each other based on environmental proximity, which incidentally make them similar in the phenotype of interest (panel D). Social stratification has been found to play a small to moderate role in partner similarity in EA[23–25]. Fourth, convergence refers to partners becoming more similar over time, either because they influence each other or because they share environments (panel E). Convergence has been found for lifestyle choices such as alcohol consumption and exercise[26]. Convergence is not a form of assortment, but an alternative explanation for partner similarity.

Each mechanism has different genetic and environmental consequences and can bias genetic[18,21] and intergenerational studies in different ways. The optimal adjustment for assortative mating depends on the underlying process, which is often unknown. Direct assortment on the observed variable is typically assumed, although several studies have found deviations from direct assortment for EA[20,21,25] and one study found deviations from direct assortment in 29 of 51 studied traits[27]. We have previously shown that adding sibling data can inform on mechanisms[20]. Our third aim was therefore to determine whether partner resemblance across a range of health phenotypes is consistent with direct assortment.

Assortative mating related to EA could be particularly important. EA relates to most health conditions and has higher partner similarity than most other traits[1,28]. Assortative mating based on EA could lead to partner similarity in health phenotypes due to indirect assortment. Assortment based on EA is also a potential explanation for cross-trait correlations between different health conditions when both conditions are related to EA. However, high EA is achieved in adulthood, potentially after meeting a partner, and may be influenced by convergence. Therefore, assortment may not take place on EA itself, but its precursors. We here additionally use grade point average (GPA) at age 16 as an early precursor of EA. Our fourth aim was to determine to what degree assortment on health phenotypes is indirect via assortment on EA or its early indicator, GPA.

In summary, we analysed educational and medical records for the parents of all first-born children born between 2016 and 2020 to parents living in Norway. Our approach had four key advantages: Population-wide data with no participation bias, early assessment, comprehensive phenotype data from primary care, and data from siblings as well as partners. First (aim 1), we studied partner similarity in education, mental, and somatic health conditions. To limit the role of convergence, we observed health 10–5 years before couples had their first child. We found positive partner correlations for all conditions, which were higher for mental than for somatic health conditions. Correlations were higher in established couples, indicating convergence. Second (aim 2), we found widespread cross-trait assortment. Education in one partner was linked with mental health in the other partner and assortment across different mental health conditions were widespread. Third (aim 3), by analysing correlations among siblings and siblings-in-law, we found frequent deviations from direct assortment. As we deal with convergence by design, deviations from direct assortment must be due to either indirect assortment or social stratification. Fourth (aim 4), we explored whether partner resemblance in health could be explained by assortment on GPA or EA. Adjusting for both partners' GPA or EA reduced correlations in health with 30–40%.

## Results
### Descriptive statistics
Our study was based on the complete Population Register of Norway. We defined as partners all pairs of opposite-sex individuals registered as parents for the first time between 2016 and 2020 (187,926 individuals in 93,963 couples). We examined 10 mental and 10 somatic health conditions in primary care records. These were measured 5–10 years before a couple had their first child to minimize effects of convergence. Women were on average 19.61 and men 21.96 years old at the start of the observational period. GPA was observed at age 16 and EA at age 30 or in 2020 for younger individuals.

Table 1 presents the prevalence of the mental and somatic health conditions, as well as conditional prevalence rates among relatives of affected individuals. Partners, siblings, and in-laws of affected individuals generally had heightened risks of having the same conditions. Figure 2 illustrates these prevalence rates among females and males with unaffected and affected partners. Those with an affected partner were more likely to have the condition themselves, although the strength of this association varied considerably by condition. The within-individual correlations for the educational outcomes and the 20 health conditions are presented in Supplemental Fig. S1. Mental health conditions exhibited stronger inter-correlations and also demonstrated larger associations with education than did the somatic health conditions. Within-person associations from logistic regression analyses are presented as odds ratios in Supplemental Fig. S2.

### Partner correlations within mental and somatic health conditions (aim 1)
Figure 3 shows in dark blue the partner correlations in educational outcomes and 20 health phenotypes (10 mental and 10 somatic)

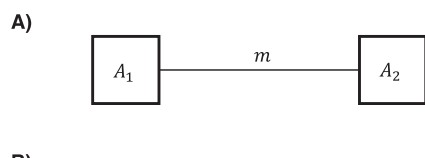

**Direct assortment**

Direct assortment, also referred to as primary phenotypic assortment, refers to partner similarity ($m$) due to selection conditional on the observed trait. In this paper, we test for deviations from direct assortment.

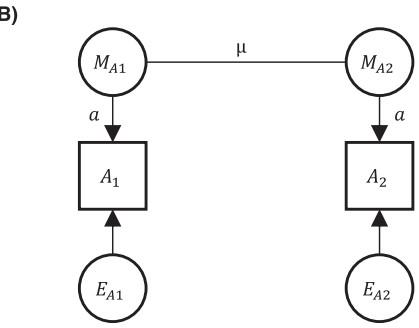

**Indirect assortment based on unknown trait**

Indirect assortment, also referred to as secondary assortment, refers to the situation that partners resemble each other in a focal trait ($A$) due to assortment in correlated traits. The identity of the correlated traits may or may not be known to the researcher. When it is not known, it may be treated as a latent variable, and its role be deduced from correlational systems. Assortment may take place directly on an unidentified or composite phenotype $M_A$, but the researcher only observes $A$. When the assorted phenotype is unknown, variance in a trait can be separated into mated ($M_A$) and unmated ($E$) variance, and the association between $M_A$ and $A$ can be estimated. This requires additional information, such as data on siblings-in-law or co-siblings-in-law. If measurement error is not taken care of, assortment on a phenotype measured with error is indistinguishable from indirect assortment based on an unknown trait. For example, assortment may be direct for depression, but indirect for a low-quality screening instrument. In this paper, indirect assortment is one of two possible explanations of deviations from direct assortment.

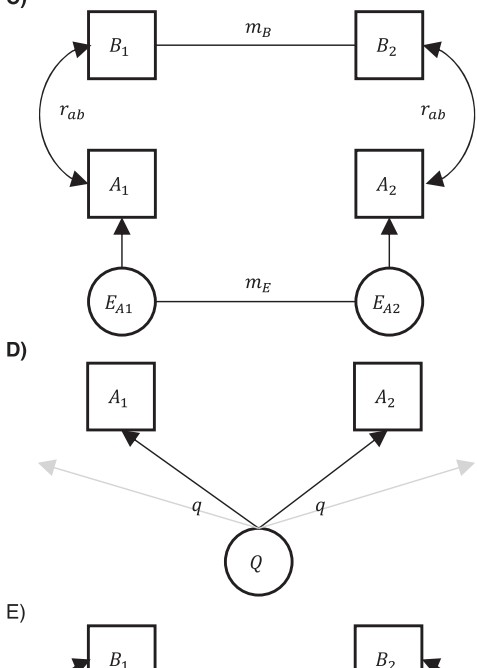

**Indirect assortment based on known trait**

With information on traits correlated with the focal trait, the degree of indirect assortment can be explicitly modelled. In the depiction, assortment on trait $A$ is thought to be secondary to assortment on trait $B$. The assignment of traits as $A$ and $B$ depends on theory, as the model cannot statistically distinguish between them. For example, partners may be similar in their level of depression due to assortment on neuroticism. The residuals $E$ can be correlated if assortment on $B$ does not fully explain partner similarity in $A$. The residual mating ($m_E$) is an upper bound estimate of direct assortment because indirect assortment may also take place on additional traits. In this paper, we test to what degree similarity in health can be secondary to assortment on education.

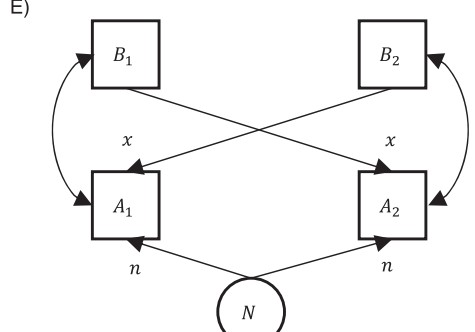

**Social stratification**

Social stratification ($Q$) refers to partner selection taking place within constrained sections of the population. For example, regions could have varying levels of educational attainment, without the education itself influencing partner choice. This is the non-genetic equivalent to population stratification and is one of the meanings of the term "social homogamy" in the literature. In this paper, social stratification is one of two possible explanations of deviations from direct assortment.

**Convergence**

Convergence refers to partners being more similar at the time of observation than at the time of couple formation. Convergence is not a form of assortment, but alternatively (or additionally) lead to partner resemblance. Two distinct processes can lead to convergence. Partners could mutually influence each other ($x$), for example if good mental health in one partner is positive for the mental health of the other partner. Furthermore, partners could share environments and experiences ($N$), leading to greater similarity, for example shared living conditions, economic conditions, or the birth of a child. In this paper, we deal with convergence by design, that is, by observing couples 5-10 years before they had their first child.

**Fig. 1 | Mechanisms of partner similarity.** Conceptual representations of mechanisms of similarity between partners 1 and 2 in trait A. **A** illustrates direct assortment. **B** illustrates indirect assortment based on unknown traits. **C** illustrates indirect assortment based on known traits. **D** illustrates social stratification. **E** illustrates convergence. Several mechanisms could co-exist, and the list does not include complex multivariate assortment.

observed 5 to 10 years prior to the couple's first child. This prospective analysis revealed positive partner correlations for all the included traits, ranging from 0.02 (allergic rhinitis) to 0.56 (substance use disorder). Notably, all the mental health conditions had higher partner correlations than all the somatic health conditions. The median partner correlation was 0.14 for mental health conditions and 0.04 for somatic health conditions. GPA correlated 0.43 and EA correlated 0.47 between partners.

## Cross-sectional assessment yields higher within-trait correlations than prospective assessment

To address the potential impact of convergence, we also conducted a cross-sectional analysis of the 10 mental and 10 somatic health conditions from 2015 to 2019, when the couples were likely already established. These cross-sectional analyses disregard the timing of childbirth. Figure 3 contrasts these cross-sectional correlations (shown in red) with the prospective correlations. The correlations between

**Table 1 | List of ICPC-2 codes for the mental and somatic health condition, prevalence in the sample (including education), and prevalence among partners and relatives of affected individuals, 10–5 years before the birth of the first child (n = 187,926 focal individuals; n = 156,335 siblings; n = 156,335 in-laws)**

| Variable | ICPC-2 codes | Index | | Partner of affected | | Sibling of affected | | In-law of affected | |
|---|---|---|---|---|---|---|---|---|---|
| | | n | % | n | % | n | % | n | % |
| University education | | 93,303 | 49.84 | 64,110 | 68.89 | 48,682 | 61.23 | 46,001 | 58.16 |
| Grade Point Average among top 20% | | 26,824 | 20.15 | 8,058 | 37.43 | 7,654 | 42.38 | 5,085 | 32.38 |
| Substance use disorders | P18, P19 | 3006 | 1.60 | 524 | 17.43 | 184 | 8.75 | 127 | 5.76 |
| Hyperkinetic disorder | P81 | 4840 | 2.58 | 412 | 8.51 | 518 | 14.67 | 157 | 4.25 |
| Personality disorder | P80 | 877 | 0.47 | 20 | 2.28 | 13 | 2.03 | <10 | |
| Psychotic disorders | P72, P98, P73 | 1420 | 0.76 | 44 | 3.10 | 47 | 4.22 | 18 | 1.58 |
| Depressive disorder | P76 | 17,086 | 9.09 | 2378 | 13.92 | 2098 | 15.92 | 1457 | 10.63 |
| Acute stress reaction | P02 | 14,288 | 7.60 | 1512 | 10.58 | 1617 | 14.59 | 978 | 8.46 |
| Sleep disturbance | P06 | 10,109 | 5.38 | 834 | 8.25 | 698 | 8.87 | 484 | 5.95 |
| Anxiety disorder | P74 | 6584 | 3.50 | 372 | 5.65 | 399 | 7.84 | 218 | 4.15 |
| Alcohol use disorders | P15, P16 | 3344 | 1.78 | 110 | 3.29 | 82 | 3.18 | 53 | 1.99 |
| Phobia/compulsive disorder | P79 | 2697 | 1.44 | 66 | 2.45 | 69 | 3.20 | 39 | 1.78 |
| Neck/back symptom/complaint | L01, L02, L03 | 35,247 | 18.76 | 7620 | 21.62 | 6155 | 21.98 | 5382 | 18.71 |
| Injury musculoskeletal | L81 | 16,043 | 8.54 | 1742 | 10.86 | 1546 | 11.80 | 1219 | 9.22 |
| Fractures | L72, L73, L74, L75, L76 | 13,489 | 7.18 | 1188 | 8.81 | 1007 | 9.10 | 823 | 7.46 |
| Naevus/mole | S82 | 20,382 | 10.85 | 2,366 | 11.61 | 2591 | 15.02 | 1853 | 10.79 |
| Acne | S96 | 11,145 | 5.93 | 790 | 7.09 | 1108 | 11.80 | 563 | 6.04 |
| Laceration/cut | S18 | 19,170 | 10.20 | 1916 | 9.99 | 1886 | 11.81 | 1611 | 10.16 |
| Headaches | N89, N90, N01, N95 | 25,716 | 13.68 | 3226 | 12.54 | 3366 | 16.26 | 2842 | 13.40 |
| Abdominal pain/cramps general | D01 | 30,338 | 16.14 | 4354 | 14.35 | 4348 | 17.96 | 3780 | 15.16 |
| Asthma | R96 | 12,593 | 6.70 | 916 | 7.27 | 1422 | 13.87 | 742 | 7.11 |
| Allergic rhinitis | R97 | 18,943 | 10.08 | 2030 | 10.72 | 2455 | 15.53 | 1542 | 9.72 |

ICPC-2 International classification of primary care-2.

partners' mental health conditions increased notably from prospective analyses, where the median correlation was 0.14, to cross-sectional analyses, where the median correlation was 0.25. We tested whether the cross-sectional and prospective correlations differed by comparing to models: one where the two correlations were estimated freely for each health condition, and another model where they were constrained to be equal. The prospective and cross-sectional correlations could not be to be equal for the 10 mental health conditions ($-2\Delta LL = 211.40$, $\Delta df = 10$, $p < 1.00e-99$). For somatic health conditions, the increases were more modest, with a median increase from 0.04 to 0.06, but the correlations could not be set to be equal ($-2\Delta LL = 63.05$, $\Delta df = 10$, $p = 9.55e-10$). Supplemental Tables S2–S3 and Supplemental Figs. S4–S9 provide complete results for the cross-sectional assessment.

**Partner correlations across different mental and somatic health conditions (aim 2)**
Returning to the prospective analyses, we investigated partner correlations across educational, mental, and somatic phenotypes. We found that partner correlations were ubiquitous across different mental health conditions. Figure 4 illustrates the partners' correlations within and across all 22 phenotypes, whereas Table 2 summarises median correlations for different categories of phenotypes. EA and GPA correlated 0.66 within individuals, implying that 56% of the variance in EA was not shared with GPA. Yet, the associations of either GPA or EA with health conditions were remarkably similar. Higher GPA or EA was generally associated with a lower risk of most health conditions in the partner, except for acne, allergic rhinitis, and naevus/mole which had negligible associations in the other direction. These conditions were

also related to higher education or better grades within individuals (see Supplemental Fig. S1).

The median correlation between education and mental health conditions in the partner was −0.16 to −0.17, depending on sex and the educational outcome. All mental health conditions were associated with all other mental health conditions in the partner, indicating widespread cross-trait assortment. The median partner correlation across different mental health conditions was 0.13, close to the within-phenotype correlation of 0.14. In contrast, most somatic conditions showed little to no relation to mental health conditions in partners, with a few exceptions (median correlation 0.03), and the cross-trait correlation for somatic conditions was minimal with a median at 0.01. Table 2 presents median correlations within and between different phenotype categories for the cross-sectional analyses. In the cross-sectional analyses, most cross-trait correlations were marginally higher, between 0.01 to 0.03, compared to the prospective analyses. We did not observe any noteworthy differences in the correlations between male and female traits. Supplemental Fig. S3 presents the within and across-trait associations as odds ratios. Results were similar to the correlation analyses and indicated widespread cross-trait associations for educational outcomes and mental health conditions.

**Testing if associations are in line with direct assortment (aim 3)**
We then explored whether the partner correlations were in line with direct assortment on the observed traits. Under direct assortment, the correlation between indirectly related individuals, such as siblings-in-law, should equate the product of the correlations that connect them, in this case, partners and siblings. Among the 187,926 parents in our sample, 156,335 had a sibling, resulting in an equal number of sibling-

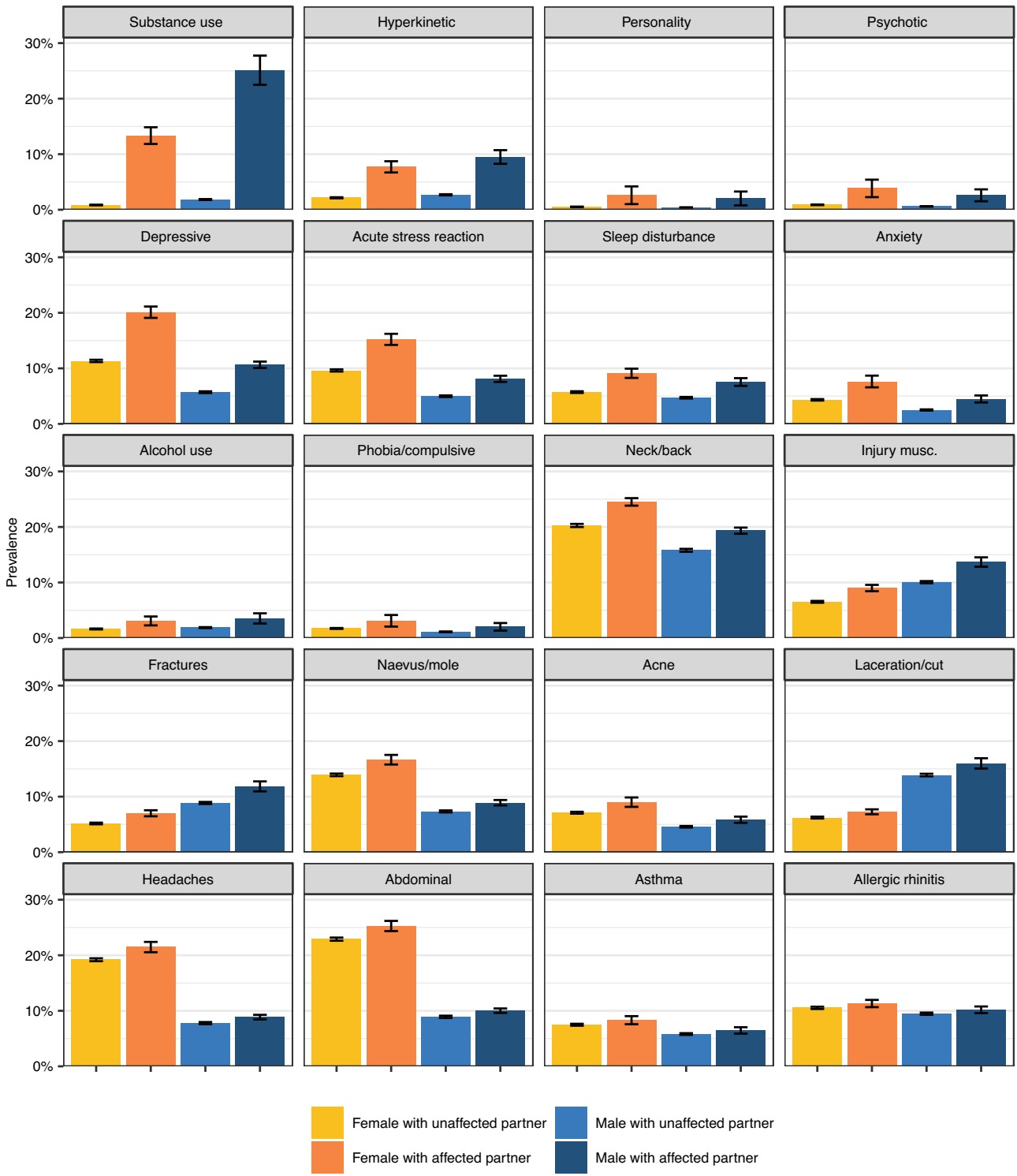

**Fig. 2 | Prevalence rates by partner health.** Prevalence of 10 mental and 10 somatic health conditions among males (*n* = 93,963) and females (*n* = 93,963) with unaffected and affected partners, 10 to 5 years before a couple had their first child. Data are presented as the proportions of diagnosed individuals, with error bars indicating 95% confidence intervals. Source data are provided in the source data file.

in-law observations. The probability of having university education depended not only on the partner's education but also the partner's sibling's educational level. When the partner did not have university education, 44.5% had university education if the partner's sibling had university education, but only 28.1% if the partner's sibling did not. Conversely, when the partner did have university education, these numbers were 75.5% and 60.6%, depending on the partner's sibling's education.

We statistically tested whether siblings-in-law correlation matched the expectation under direct assortment. Table 3 presents the correlations for partners, siblings, and siblings-in-law in all the 22 traits, as well as the 'in-law inflation factor', which compares the observed in-

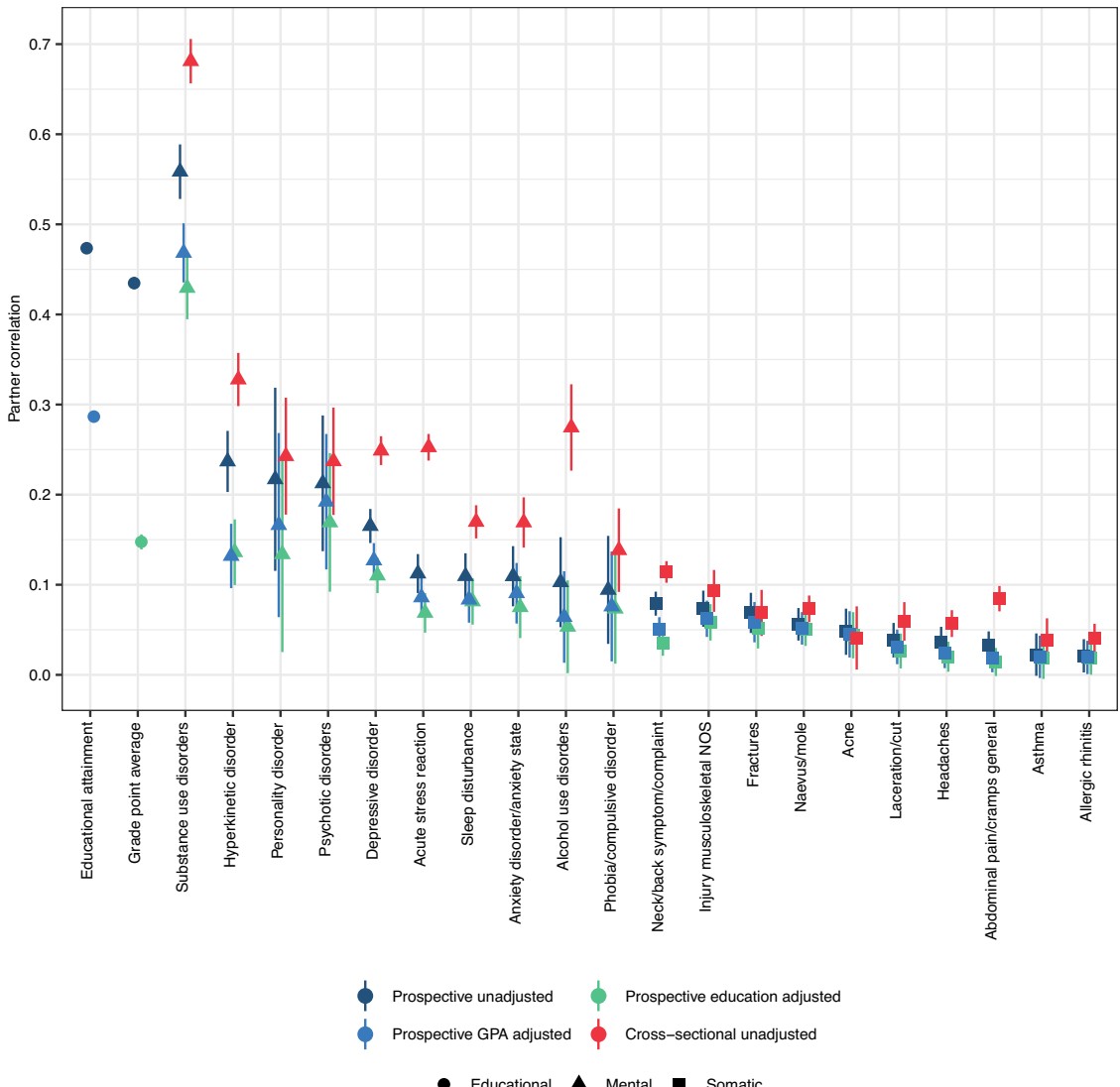

**Fig. 3 | Within-trait partner correlations with various adjustments.** Correlations between female and male partners ($n = 93,963$ couples) for educational outcomes and 10 mental health and 10 somatic health phenotypes 10 to 5 years before they had their first child and cross-sectionally in 2015–2019. Error bars indicate 95% confidence intervals. Source data are provided in the source data file.

law correlations to those predicted under direct assortment. It was calculated by dividing the observed correlations between siblings-in-law by the product of the sibling and partner correlations. This was above 1.00 for 20 of 22 phenotypes, indicating that direct assortment cannot account for the observed correlations. False discovery rate adjusted p-values indicated statistically significant deviations from direct assortment at the $\alpha = 0.05$ level for GPA, EA, 3 mental health conditions, and 4 somatic health conditions. Logistic regression presented in Supplementary Table S1 indicated independent associations with siblings-in-law for the two educational outcomes, 5 mental and 4 somatic health conditions, after accounting for partner associations. This concurs with deviations from direct assortment.

### Indirect assortment on health via assortment on educational attainment (aim 4)

We proceeded to test whether assortment on GPA or EA could drive partner similarity in health. Figure 3 shows the residual partner correlations after adjustments for GPA (in base blue) or EA (in green). The partner correlation in EA adjusted for GPA was twice as strong ($r = 0.29$) as the partner correlation in GPA adjusted for EA ($r = 0.15$),

suggesting that EA is more strongly related to assortment than GPA is. For mental disorders, the median partner correlations were reduced from 0.14 to 0.11 after adjustment for GPA (down 21.6%) and to 0.10 after EA adjustment (down 31.0%). The median partner correlation for somatic disorders was already low at 0.04 and remained at 0.04 (down 12.0%) after adjustment for GPA and was reduced to 0.03 (down 29.1%) after adjustment for EA.

Assortment on GPA or EA could also influence cross-trait partner correlations. Table 2 summarizes the median correlations and Figs. 5 and 6 provide the complete correlation matrices after adjustment for GPA and EA, respectively. The median partner correlation between different mental health conditions was reduced from 0.13 to 0.08 (down 32.2%) after adjustment for GPA and to 0.07 after adjustment for EA (down 41.1%). The median partner correlation between different somatic disorders was stable at 0.01.

### Discussion

Studying the complete set of first-time Norwegian parents, we found positive partner correlations in GPA, EA, and all analysed mental and somatic health conditions, observed from 10 to 5 years before the birth

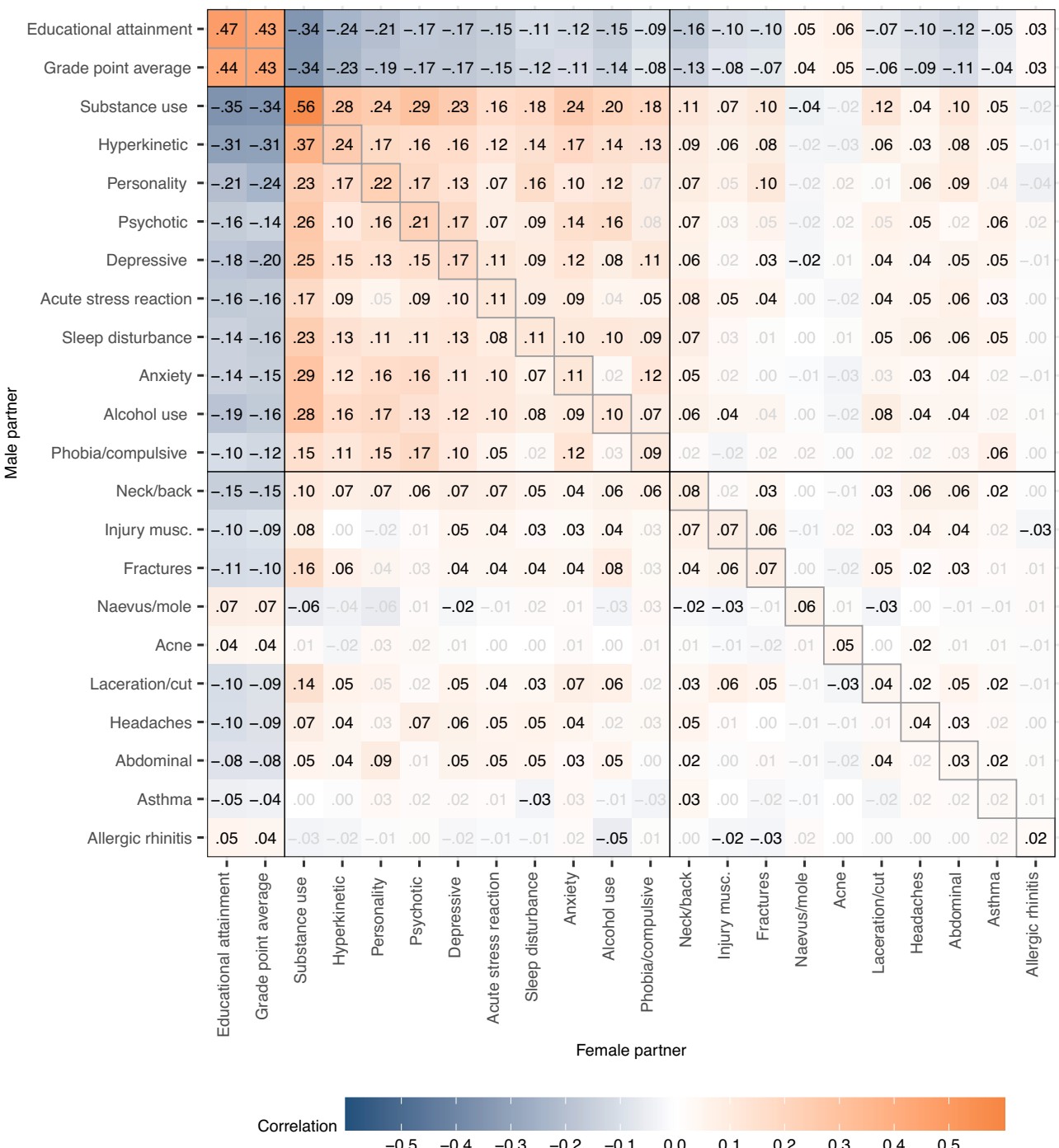

**Fig. 4 | Prospective partner correlations.** Within and across-trait partner correlations for educational outcomes, 10 mental health conditions, and 10 somatic health conditions, 10 to 5 years before first child (*n* = 93,963 couples). Adjusted for age. We tested whether the correlations differ from zero using two-sided *z*-tests based on the estimated correlations and their standard errors provided by OpenMx. Significant correlations (*p* < 0.05 after Benjamini-Hochberg adjustment) are shown in black. Exact *p*-values are provided in the Source Data file.

of a couple's first child. The initial similarity and later convergence were larger for mental than somatic health conditions. We also observed ubiquitous cross-trait correlations for mental health conditions, which in prospective analyses were approximately as large as the within-trait correlations. The pattern of correlations between relatives indicated deviations from direct assortment on several of the observed phenotypes. Although partner correlations could be partially explained as by-products of assortment related to education, this cannot be a primary explanation of partner correlations in mental health.

## Mental health in early adulthood associated with partner selection

Our study expands on previous research by including the whole population, studying diagnosed health conditions, and contrasting the importance of mental versus somatic health conditions. To

**Table 2 | Median correlations (r) and median standard errors (SE) within and across traits in different categories (n = 93,963 couples)**

| Trait in female | Trait in male | Within or across traits | Prospective, adjusted for age | | Prospective, adjusted for age GPA | | Prospective, adjusted for age and educational attainment | | Cross-sectional, adjusted for age | |
|---|---|---|---|---|---|---|---|---|---|---|
| | | | Median r | Median SE | Median r | Median SE | Median r | Median SE | Median r | Median SE |
| EA | EA | Within | 0.47 | 0.00 | 0.29 | 0.00 | – | – | 0.47 | 0.00 |
| EA | GPA | Across | 0.44 | 0.00 | – | – | – | – | 0.44 | 0.00 |
| GPA | EA | Across | 0.43 | 0.00 | – | – | – | – | 0.43 | 0.00 |
| GPA | GPA | Within | 0.43 | 0.00 | – | – | 0.15 | 0 | 0.43 | 0.00 |
| EA | Mental | Across | −0.17 | 0.01 | −0.09 | 0.01 | – | – | −0.19 | 0.01 |
| Mental | EA | Across | −0.16 | 0.01 | −0.08 | 0.01 | – | – | −0.17 | 0.01 |
| GPA | Mental | Across | −0.16 | 0.01 | – | – | −0.05 | 0.01 | −0.19 | 0.01 |
| Mental | GPA | Across | −0.16 | 0.01 | – | – | −0.03 | 0.01 | −0.16 | 0.01 |
| EA | Somatic | Across | −0.09 | 0.01 | −0.04 | 0.01 | – | – | −0.10 | 0.01 |
| Somatic | EA | Across | −0.08 | 0.01 | −0.04 | 0.01 | – | – | −0.10 | 0.01 |
| GPA | Somatic | Across | −0.08 | 0.01 | – | – | −0.01 | 0.01 | −0.10 | 0.01 |
| Somatic | GPA | Across | −0.06 | 0.01 | – | – | −0.01 | 0.01 | −0.09 | 0.01 |
| Mental | Mental | Within | 0.14 | 0.02 | 0.11 | 0.02 | 0.10 | 0.02 | 0.25 | 0.01 |
| Mental | Mental | Across | 0.13 | 0.02 | 0.08 | 0.02 | 0.07 | 0.02 | 0.16 | 0.02 |
| Mental | Somatic | Across | 0.03 | 0.02 | 0.02 | 0.02 | 0.01 | 0.02 | 0.05 | 0.01 |
| Somatic | Mental | Across | 0.03 | 0.02 | 0.02 | 0.02 | 0.01 | 0.02 | 0.05 | 0.01 |
| Somatic | Somatic | Within | 0.04 | 0.01 | 0.04 | 0.01 | 0.03 | 0.01 | 0.06 | 0.01 |
| Somatic | Somatic | Across | 0.01 | 0.01 | 0.01 | 0.01 | 0.01 | 0.01 | 0.02 | 0.01 |

*GPA* grade point average, *EA* educational attainment.

minimize the influence of convergence, we examined young adults before parenthood and typically before partnership formation. We demonstrate that partners resemble each other in mental health before they are likely to have met. As far as we are aware, partner resemblance in mental health assessed before couple formation has previously only been found for self-reported symptoms in a cohort study[29]. Our prospective analyses and use of proper diagnoses indicate that there is assortment on the liability to mental disorders, as questioned by Yengo[11]. The lack of correlations between partners' polygenic indices in previous studies is likely due to limited discovery samples and small effects of each causal variant, giving the polygenic indices low predictive value for mental health conditions. An alternative explanation is that overrepresentation of healthy and well-educated individuals in cohort studies restricts the range and downwardly biases partner correlations. For example, we observed a partner correlation of 0.48 for EA, compared to 0.42 in a Norwegian cohort[20]. However, for mental health, our estimates of correlations between partners-to-be were slightly lower (median r = 0.13) than in a cohort study assessing global mental health among future partners (r = 0.16)[29]. Our study indicated that mental health conditions were more strongly related to partner selection than somatic health conditions common in young adulthood. This is not surprising, given that mental health is linked with marriage and fertility[30] and could indicate desirability to potential partners.

Partner correlations in mental health were considerably higher at the end than at the start of the observational period. This highlights that studies on established couples can typically only inform on correlations, and that convergence needs to be addressed before interpreting correlations as indicative of assortment[2,16]. This increase does not necessarily reflect mutual influences or shared experiences; it could also be that partner selection is based on vulnerabilities to mental disorders that manifest as diagnosable conditions later in life (indirect assortment). We observed a change in resemblance from 10 to 5 years before childbirth until the years surrounding childbirth in the same couples—the increased resemblance may be even more pronounced among older couples.

**Assortment across mental health conditions is ubiquitous**

The cross-trait correlations for different mental health conditions in the two partners were almost as strong as within-trait correlations (median r = 0.14 vs 0.13). Hence, individuals tend to mate with partners who share similarly good or poor mental health, with the specific type of health condition being subordinate. Such results align with assortment on perceived attractiveness, itself influenced by both mental and educational traits. Thus, the partner correlations observed across different traits likely reflect indirect assortment. Our results differ from a study that found assortment primarily on symptoms of specific disorders[8]. However, that study used data on established couples, which, according to our results, have increased in within-trait correlations.

The positive manifold across mental health conditions in partners can in the next generation increase genetic correlations between traits; not because the same set of genes are associated with different traits, but because genetic liabilities to different traits co-occur in the same individuals[4]. This can contribute to the frequently observed "p-factor"[31]. In addition, cross-trait assortment can easily lead to bias in genetic studies, as unmeasured genetic variants can be related to measured variants as well as the outcome of interest. This can inflate estimates in genome-wide association studies and violate the exclusion criteria in Mendelian randomization studies[19]. In the presence of cross-trait assortment, the results of such studies should be interpreted with caution.

Individuals with better grades or higher education were less likely to have partners with mental and somatic health conditions. This suggests a trade-off between different attractive traits in partners, indicating competition for healthy partners rather than matching on similarity. Nevertheless, the remarkably high correlation for substance use disorder could indicate genuinely different lifestyle preferences. Modelling of variations in preferences may be vital to fully understand cross-trait assortment[3,32]. There was little assortment across different somatic conditions or across mental and somatic conditions. Still, most correlations were positive, and mostly so among those involving various types of pain, possibly reflecting the mental aspect of pain.

**Table 3 | Correlations between relatives in educational outcomes and 10 mental and 10 somatic health conditions 10 to 5 years before a couple had their first child, including 95% confidence intervals, along with tests of deviations from direct assortment**

| Variable | r (partner) | r (sibling) | r (inlaw) | Inlaw inflation factor (IIF)[a] | Deviation from direct assortment, p-value[b] |
|---|---|---|---|---|---|
| Educational attainment | 0.48 [0.47, 0.48] | 0.40 [0.40, 0.40] | 0.29 [0.28, 0.29] | 1.50 | <1.00e-99 |
| Grade point average | 0.42 [0.42, 0.43] | 0.52 [0.52, 0.53] | 0.29 [0.29, 0.30] | 1.33 | <1.00e-99 |
| Substance use disorders | 0.54 [0.51, 0.57] | 0.37 [0.34, 0.40] | 0.27 [0.23, 0.30] | 1.32 | 0.902 |
| Hyperkinetic disorder | 0.24 [0.20, 0.27] | 0.43 [0.40, 0.45] | 0.10 [0.07, 0.13] | 0.97 | 0.279 |
| Personality disorder | 0.21 [0.11, 0.32] | 0.18 [0.09, 0.26] | 0.10 [0.00, 0.20] | 2.70 | 0.902 |
| Psychotic disorders | 0.21 [0.13, 0.28] | 0.21 [0.16, 0.26] | 0.04 [−0.04, 0.11] | 0.84 | 1.78e-04 |
| Depressive disorder | 0.16 [0.15, 0.18] | 0.23 [0.21, 0.24] | 0.07 [0.06, 0.09] | 1.91 | 0.015 |
| Acute stress reaction | 0.11 [0.09, 0.13] | 0.25 [0.24, 0.27] | 0.05 [0.04, 0.07] | 1.93 | 0.112 |
| Sleep disturbance | 0.11 [0.08, 0.13] | 0.15 [0.13, 0.17] | 0.04 [0.02, 0.06] | 2.34 | 0.279 |
| Anxiety disorder | 0.11 [0.07, 0.14] | 0.20 [0.17, 0.22] | 0.04 [0.01, 0.07] | 1.85 | 0.235 |
| Alcohol use disorders | 0.10 [0.06, 0.15] | 0.14 [0.10, 0.18] | 0.05 [0.00, 0.10] | 3.43 | 0.554 |
| Phobia/compulsive disorder | 0.09 [0.03, 0.15] | 0.13 [0.09, 0.17] | 0.03 [−0.02, 0.08] | 2.64 | 1.60e-09 |
| Neck/back symptom | 0.08 [0.07, 0.09] | 0.13 [0.12, 0.14] | 0.05 [0.04, 0.06] | 4.34 | 1.68e-04 |
| Injury musculoskeletal | 0.08 [0.06, 0.09] | 0.13 [0.12, 0.15] | 0.05 [0.03, 0.06] | 4.61 | 0.033 |
| Fractures | 0.07 [0.05, 0.09] | 0.09 [0.07, 0.11] | 0.03 [0.01, 0.05] | 4.61 | 0.902 |
| Naevus/mole | 0.06 [0.04, 0.07] | 0.14 [0.13, 0.16] | 0.01 [−0.01, 0.02] | 1.11 | 0.235 |
| Acne | 0.04 [0.02, 0.07] | 0.21 [0.20, 0.23] | 0.02 [0.00, 0.04] | 2.66 | 0.162 |
| Laceration/cut | 0.04 [0.02, 0.06] | 0.07 [0.06, 0.09] | 0.02 [0.00, 0.03] | 5.76 | 0.003 |
| Headaches | 0.04 [0.02, 0.06] | 0.11 [0.10, 0.13] | 0.03 [0.01, 0.04] | 5.90 | 0.003 |
| Abdominal pain | 0.03 [0.02, 0.05] | 0.10 [0.09, 0.12] | 0.02 [0.01, 0.04] | 6.92 | 0.235 |
| Asthma | 0.02 [0.00, 0.04] | 0.23 [0.21, 0.24] | 0.02 [−0.00, 0.04] | 3.92 | 0.235 |
| Allergic rhinitis | 0.02 [0.00, 0.04] | 0.19 [0.17, 0.20] | 0.01 [−0.00, 0.03] | 3.79 | 0.166 |

Adjusted for sex and year of birth (n = 93,963 couples; n = 156,335 siblings; n = 156,335 in-laws). Source data are provided in the Source data file.
[a]IIF is expected to equal 1.00 under direct assortment. Deviations from 1.00 can be due to indirect assortment and social stratification, whereas we deal with convergence by design.
[b]The p-values result from likelihood-ratio tests (1 degree of freedom) comparing a constrained model, where the in-law correlation is the product of partner and sibling correlations, to an unconstrained model with independent estimates for each relationship type. The p-values are adjusted for false discovery rate using the Benjamini-Hochberg method. A low p-value signifies a poor fit for direct assortment.

Regardless of genetic consequences, the widespread cross-trait assortment could enhance negative consequences for children, as they may be influenced by both low education and poor mental health in their parents[33,34].

**Partner correlations are generally inconsistent with direct assortment**

When accounting for assortative mating to avoid bias, studies make assumptions about the mechanisms involved. Typically, they assume direct assortment on the observed variables[35,36]. Our results challenge this notion. The siblings-in-law correlations exceeded those expected under direct assortment, suggesting that direct assortment is not a sufficient explanation for partner resemblance and that studies relying on this assumption can be biased. Deviations from direct assortment have previously been reported for EA[20,24,25,37]. We extended this observation to GPA and a range of health conditions. Our results align with another study that observed deviations from direct assortment in 29 of 51 traits[27], mainly different traits than those studied here.

Although the phenotypic model could be falsified, the underlying mechanisms remain elusive. Both indirect assortment and social stratification[38] could increase in-law correlations disproportionately and explain our observations. In any case, partner resemblance is not solely due to assortment based on the observed phenotypes. Whether parts of the partner correlations in mental health are due to causal influences on partner choice remains to be determined. Identifying the traits that actively determine assortment is an important question for future studies. It might be more strongly related to general vulnerability to psychopathology[31] than to specific disorders. Due to the strong cross-trait assortment, such causal effects may be more plausible at the level of general mental health, rather than for specific

diagnoses. A previous study indicated that partner similarity in many traits was driven by assortment on a few key traits[38], but it did not include mental disorders. Future studies may explore whether partner resemblance across many traits can be more parsimoniously explained by assortment on one or a small number of dimensions.

Indirect assortment need not be based on symmetric assortment on a manifest phenotype. Measurement error can be indistinguishable from indirect assortment on an unknown trait. Assortment may then be said to be direct for the true values of a trait, but indirect for an imperfect indicator. As measurement error is widespread and relatively easy to estimate, accounting for measurement error could improve future studies on assortment. Indirect assortment could also be related to impression management, whereby partner selection could take place on successful misrepresentations of one's characteristics. This should, however, not influence sibling correlations. Finally, correlations in trait preferences among siblings can increase correlations between distant affines, such as co-siblings-in-law[32]. Hence, models of preferences may be needed to fully understand similarities in wider family networks.

Assortment leads to correlations between all genetic and environmental influences in one partner and those in the other. When parental traits leave a mark on their children through vertical transmission, this assortment leads to an intertwining of genetics and environment in the children. This can substantially increase gene-environment correlations in the child generation, which again increases the genetic similarity between partners[9] (formula S1.8). If there is indirect assortment, the partner similarity in assorted factors will be larger than indicated by the observed variables, and the intergenerational consequences can be underestimated. Intergenerational studies therefore need to carefully model indirect assortment. Regardless of

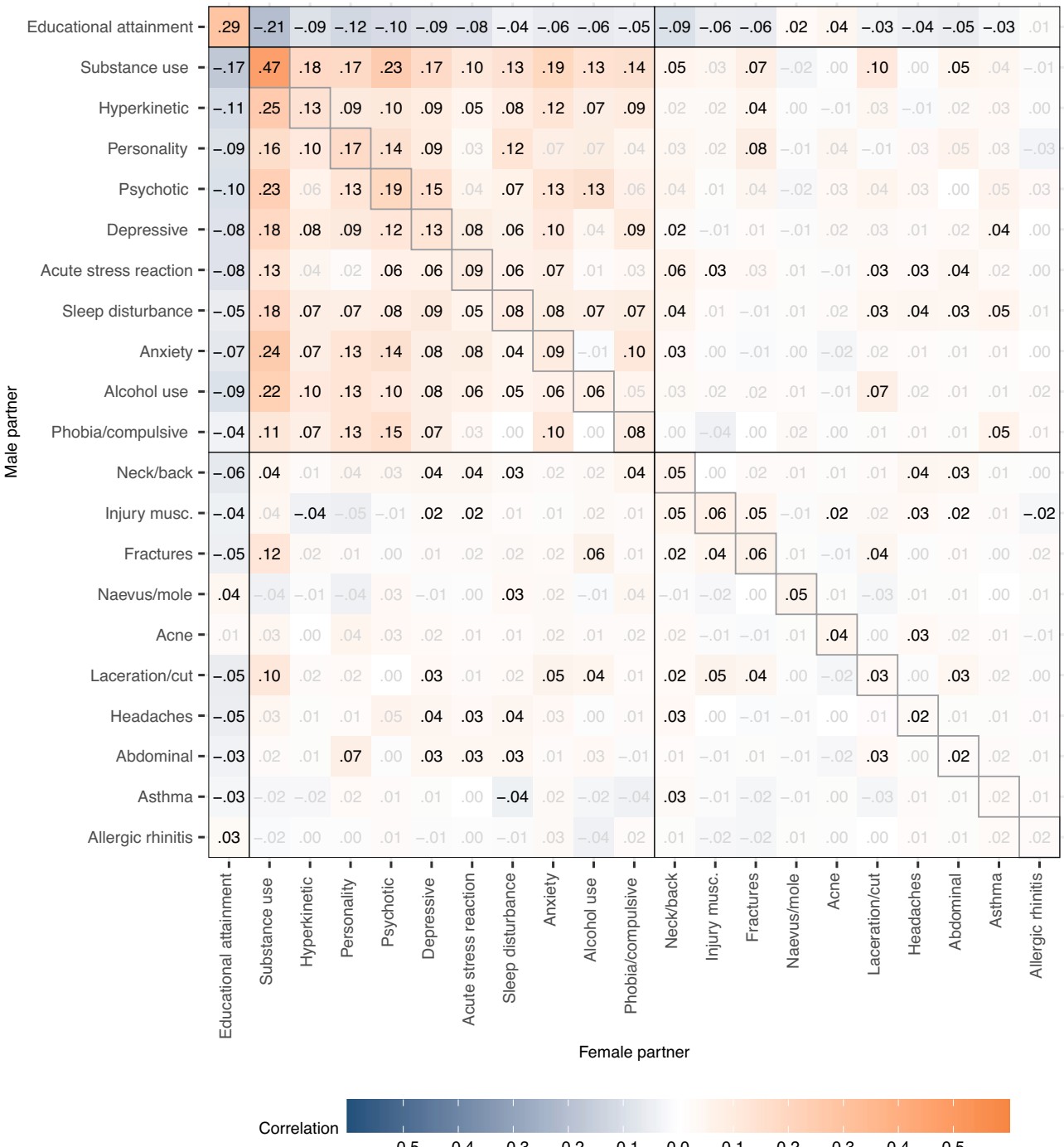

**Fig. 5 | Prospective partner correlations adjusted for grade point average.**
Within and across-trait partner correlations for 10 mental health conditions, and 10 somatic health conditions, 10 to 5 years before first child (*n* = 93,963 couples). Adjusted for age and grade point average. Correlations shown in black have *p*-values < 0.05 after adjusting for the false discovery rate. We tested whether the correlations differ from zero using two-sided *z*-tests based on the estimated correlations and their standard errors provided by OpenMx. Significant correlations (*p* < 0.05 after Benjamini-Hochberg adjustment) are shown in black. Exact *p*-values are provided in the Source Data file.

mechanism and possible genetic consequences of assortative mating[18], the potential social consequences of partnership composition could remain.

### Assortment on educational attainment partially explains health similarity

Given the known correlation between education and health status, one should expect a partner with higher education to, on average, also enjoy better health. Indeed, when we adjusted for both partners' GPA or EA, partner correlations within and across mental disorders were reduced. Hence, similarities in mental health could to some degree be by-products of on education or its precursors. Nevertheless, correlations within and across mental disorders remained significant, indicating that these were not solely by-products of assortment based on education. Hence, mental health is related to partner selection independently of observed education. Partner correlations within and across different somatic health conditions were close to zero both before and after these adjustments.

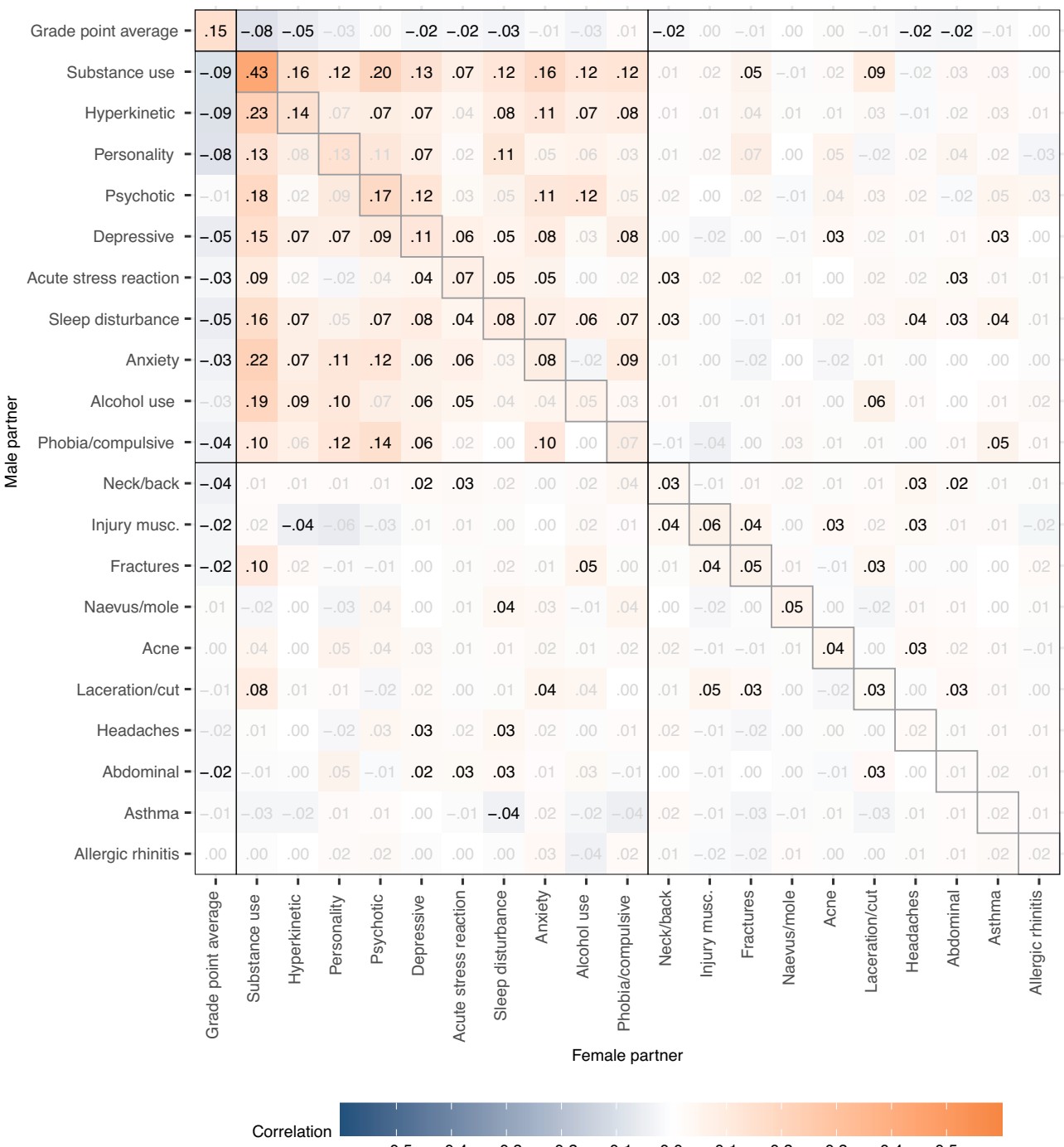

**Fig. 6 | Prospective partner correlations adjusted for grade educational attainment.** Within and across-trait partner correlations for 10 mental health conditions, and 10 somatic health conditions, 10 to 5 years before first child (*n* = 93,963 couples). Adjusted for age and educational attainment. Correlations shown in black have *p*-values < 0.05 after adjusting for the false discovery rate. We tested whether the correlations differ from zero using two-sided *z*-tests based on the estimated correlations and their standard errors provided by OpenMx. Significant correlations (*p* < 0.05 after Benjamini-Hochberg adjustment) are shown in black. Exact *p*-values are provided in the Source Data file.

It must be noted that EA was not measured prospectively; at the young age of approximately 20 years, many individuals are yet to obtain their highest education. Individuals can select partners based on the traits that exist at this age and that lead to later EA, in which case the adjustment is defendable. However, it is also possible that the adjustment for EA is an overadjustment because one's own or the partner's health could influence education. Using GPA as an alternative indicator of educational potential reduces this issue because it is typically achieved before partners meet. However, each partner's mental health could have influenced their own GPA, meaning that the true assortment on mental disorders could in principle be slightly larger than indicated by our study. GPA was somewhat less strongly linked to the partner's health than EA was. This could suggest that traits that influence EA are more important for mate choice than traits influencing GPA. Interestingly, siblings were more similar in GPA than EA, but this was reversed in partners. Cognitive abilities and conscientiousness influence both GPA and EA, but there could be differences in ambitions, achieved status, or social background. Roughly

half of the variance in EA was not shared with GPA, indicating that there are important differences between the two.

The current study indicates, as do also previous studies[20,25], that the strong partner resemblance in EA is due to even stronger resemblance in an unobserved factor. This was also the case for GPA. Assortment for EA and GPA was itself indirect; therefore, unidentified factors must exist that contribute to partner similarity in education as well as in health. This aligns with a previous study that chain-linked in-laws and inferred far greater partner similarity in latent (unidentified) advantages than in the observed level of education[25]. Our inclusion of GPA early in life is novel, however, it did not capture these latent factors any better than EA. Future research could try to identify traits that account for the sorting process and understand how they relate to partner similarity across observed traits. This may include social status more broadly defined or health in childhood.

## Limitations
This study has some limitations that one should consider when interpreting the findings. First, the medical records are proxies for actual health conditions, as not all individuals with health issues seek medical care. This prevented the study of conditions below the threshold of medical attention. This issue is reduced as the tetrachoric correlations model these thresholds. Also, our use of primary care data captures a larger proportion of cases than specialist care data alone[39], which has been used in previous studies[2]. This further mitigates potential biases. We could only study somatic conditions that were common among parents-to-be in young adulthood. The results are not necessarily representative for other somatic health conditions; in particular, assortment on rare health conditions is unknown. This also prevented the study of health conditions with a higher average age of onset, such as cardiometabolic conditions and cancers. However, conditions that develop after couple formation cannot directly influence its composition. Second, our focus on parents of children born in Norway between 2016 and 2020 could limit the generalizability to other populations or time periods. Third, we cannot rule out that some partners had already influenced each other at the start of the observational period in early adulthood. Nevertheless, the prospective nature of our study is a major advancement over previous studies, and the comparison with cross-sectional data emphasizes the impact of this analytic decision. The gap of 5 years between the end of health observation and the birth of the first child exceeds the median duration of relationships, suggesting that most couples were unacquainted during the health observation period. Fourth, we used tetrachoric correlations, based on the assumption of an underlying normally distributed liability. Whereas this could be reasonable for mental health conditions, some somatic health conditions are binary in their nature, such as fractures. This could lead to an over-estimation of partner correlations. However, this would, if anything, make the difference between mental and somatic health conditions larger. In addition, this did not affect the tests of direct assortment (Supplemental Scripts S1–S2), and results were consistent in logistic regression.

In conclusion, this study provides evidence for assortative mating patterns across GPA, EA, and 20 health conditions, up to 10 years before partners had their first child in data without participation bias. Among the health conditions, mental health conditions were particularly strongly related to partner selection. We observed vast cross-trait assortment for mental health conditions, indicating that individuals match on overall mental health, rather than on specific health conditions. The link with education might indicate trade-offs for overall attractiveness. This questions assumptions in genetic designs and could have consequences for the distribution of risk factors among children. In general, partner resemblance could not be explained with direct assortment, however, GPA or EA could only to a moderate degree account for partner similarity in mental health. The use of prospective data ensured that partner resemblance was not merely

due to convergence, and the comparison with cross-sectional data indicates that studies without prospective data do not precisely reflect assortative mating. Indirect assortment appears the best explanation for partner similarity, raising important questions on mate choice and complicating modelling of partner similarity.

## Methods
The study was approved by The Regional Committees for Medical and Health Research Ethics, Southern and Eastern Norway (project #2018/434). The committee waived the requirement for informed consent due to the use of de-identified administrative data.

### Sample and design
The Population Register of Norway consisted of 8,589,458 individuals born between 1855 and 2020 who were alive and living in Norway after 1964. We combined this with information on publicly funded health care, available from 2006 to 2019. We defined a couple as the two registered parents of a child and studied all opposite-sex parent pairs who had their first child born between 2016 and 2020. This let to observation of 93,963 couples and 187,926 parents (93,963 males and 93,963 females). Only opposite-sex parents were included in the sample, as partner similarity in same-sex couples warrants separate studies. We included only couples who were both registered as living in Norway for the 10 to 5 years prior to the child's birth. For each parent, we drew a random full sibling. Among the 187,926 parents, 156,335 had a sibling, hence, we also had data on an equal number of pairs of siblings-in-law. In 65,902 cases, both partners had siblings.

We observed health of the parents from 10 to 5 years prior to the birth of their child. For instance, for a child born in January 2016, we observed health from January 2006 to December 2010, whereas for a child born in December 2020, we observed health from December 2010 to November 2015. The 5-year lag between health observations of parents and the child's birth was intended to limit the influence of convergence on the results, by measuring them early in the relationship. Although some parents will have known each for longer, the duration of the sexual relationships with the father before the first pregnancy had a median of 4 years (first quartile: 2 years; third quartile: 6 years) among 31,651 mothers in the Norwegian Mother, Father, and Child Cohort Study (original data analyses, a general description of the sample has been provided previously[40]). To study convergence, we additionally observed health in the last 5 years available, from 2015 to 2019 for all couples regardless of when they had their first child. This is around the time they had their first child. The mean birth year was 1988 for mothers and 1986 for fathers. Mothers were on average 29.61 and fathers 31.96 years old when they had their first child (19.61 and 21.96 at the start of the observational period).

### Measures
**Educational attainment.** Educational attainment was available in eight categories, ranging from "no education" to "Ph.D.", coded according to the Norwegian Classification of Education. We used educational attainment at age 30 or the highest educational attainment at the end of the observational period as a continuous variable after recoding it into years of completed education.

**Grade point average.** Norwegian students are evaluated at the end of compulsory education, usually the year they turn 16. The Grade Point Average (GPA) is calculated as the average of all final-year teacher-evaluated grades and externally graded exams. The GPA is used for ranking students applying for admission to upper secondary education. Students therefore have an incentive to perform well. We standardized the GPA score (mean = 0, SD = 1) within each birth year cohort to adjust for grade inflation. Even the lowest grades go into the GPA score, also those that would not be considered passing at a higher level of education. This means that nearly all students have a valid GPA. GPA

was available for individuals born in 1985 or later. In total 77.4%) of mothers ($n = 72,727$) and 64.3% of fathers ($n = 60,412$) had valid GPA scores. GPA was used as a continuous variable.

**Mental and somatic health.** All persons who legally reside in Norway are members of the National Insurance Scheme and assigned a general practitioner. General practitioners and other health service providers, such as emergency rooms, send billing information to a governmental organization along with a diagnosis or reason for the visit in order to receive reimbursements. Due to economic incentives, it is unlikely that health visits go unreported. Diagnostic information is coded according to the International Classification of Primary Care (ICPC-2)[41]. The ICPC-2 contains both diagnoses and complaints. Linkage between data sources is possible via the unique national identity number.

We analysed 20 health conditions, of which 10 were mental health conditions. These covered a broad spectre of mental health conditions, corresponded to well-known conditions, and were sufficiently common to be analysed in both sexes. These analysed conditions were Depressive disorder, Anxiety disorders, Phobia/compulsive disorder, Acute stress reaction, Sleep disturbance, Alcohol use disorders, Substance use disorders, Hyperkinetic disorder (ADHD), Psychotic disorders, and Personality disorder. Likewise, we analysed 10 somatic health conditions. They were selected for their diversity in covering different health issues and for being sufficiently prevalent in both sexes in our sample of young adults who later became parents. The included conditions were Headaches, Neck/back symptoms/complaints, Abdominal pain/cramps general, Fractures, Acne, Injury musculoskeletal, Asthma, Allergic rhinitis, Laceration/cut, Naevus/mole. If at least one entry with the code was present between 10 and 5 years before the birth of the first child, the person was defined as having the condition. The ICPC-2 codes included in each condition are listed in Table 1.

**Statistical analyses**
We first described the prevalence of the health conditions by relationship type (partners, siblings, siblings-in-law). We then calculated correlations between partners (aim 1) while adjusting for birth year. We used OpenMx 2.21.8 in R 4.1.3 to estimate the correlations using Full Information Maximum Likelihood (FIML), thereby using all available data, whether complete or incomplete. Adjustments were made by adding the definition variables with slopes to means of the models. For the binary variables (all except GPA and EA), we used a liability threshold model. Hence, we used tetrachoric correlations for associations involving binary health outcomes, polyserial correlations for the associations involving GPA or EA and binary health outcomes, and Pearson correlations for associations involving only GPA and/or EA. We then estimated associations between different phenotypes in the two partners (aim 2) in a corresponding manner.

We then calculated correlations between siblings and siblings-in-law and compared these to the partner correlations to test whether the results were consistent with direct assortment on the observed traits (aim 3). Under direct assortment, the correlation for siblings-in-law ($r_{inlaw}$) equals the product of the correlations for partners ($r_{partner}$) and siblings ($r_{sibling}$). We have elaborated on this and provided supporting simulations previously[20]. By testing, if the observed correlations adhered to this pattern, we can detect deviations from direct assortment. We define an in-law inflation factor (IIF) as the ratio of the correlation between siblings-in-law to the expectation under direct assortment:

$$\text{IIF} = \frac{r_{inlaw}}{r_{sibling} * r_{partner}} \tag{1}$$

Figure 7 displays four processes that can lead to partner similarity and allows path tracing expected correlations, and hence IIF, in the different scenarios. We here consider their implications in

isolation, although combinations of the mechanisms are possible. Direct assortment (panel A of Fig. 7) assumes that partnerships are based on the observed phenotype. By applying path tracing rules allowing for co-paths[36] to Fig. 7, one can see that $r_{partner} = m$, $r_{sibling} = r_s$, and $r_{inlaw} = mr_s$. Hence, $\text{IIF} = \frac{r_{inlaw}}{r_{partner} * r_{sibling}} = \frac{mr_s}{m * r_s} = 1$, and deviations from 1.00 are not consistent with direct assortment. For simplicity, we assume unit variance. With indirect assortment (panel B of Fig. 7), partners are similar due to assortment based on an unknown (latent) phenotype. Siblings are similar in the same latent phenotype ($r_m$) and may share additional similarities ($E = (1 - a^2)r_e$). In isolation, indirect assortment gives $r_{partner} = a^2\mu$, $r_{sibling} = a^2 r_m + E$, $r_{inlaw} = a^2\mu r_m$, and $\text{IIF} = \frac{r_{inlaw}}{r_{partner} * r_{sibling}} = \frac{a^2\mu r_m}{a^2\mu * (a^2 r_m + E)} = \frac{1}{a^2 + \frac{E}{r_m}}$. When there is not a residual correlation between siblings ($r_e = 0$), this reduces to $\text{IIF} = \frac{1}{a^2}$, which is always $\geq 1.00$ as long as $-1 < a < 1$. A large value for $E$ reduces IIF, which will be 1.00 if $r_e = r_m$, and possibly below 1.00 if $r_e > r_m$. However, we consider large values for $E$ to be rare, because it is residualized on the component of a trait that matters to other individuals through mate selection and is further reduced by measurement error. Social stratification (panel C of Fig. 7) influences all individuals to the same degree, and in isolation leads to $r_{partner} = q^2$, $r_{sibling} = q^2$, $r_{inlaw} = q^2$, and $\text{IIF} = \frac{r_{inlaw}}{r_{partner} * r_{sibling}} = \frac{q^2}{q^2 * q^2} = \frac{1}{q^2}$. Hence, $q \neq 0$ leads to an IIF above 1.00 ($|q| < 1$). Convergence (panel D of Fig. 7) can refer to two processes increasing partner similarity. Shared environments exclusively influence partner correlations ($n^2$), whereas mutual influences primarily influence partner correlations ($2xa$) while having a smaller effect on in-law correlations ($xr_m a$). Because convergence primarily influences partner correlations, it will increase the denominator and reduce IIF. We do not consider convergence here further, as we deal with by design.

In sum, an IIF above 1.00 can be explained by both social stratification and indirect assortment, although we cannot distinguish between these processes without additional data. This is a topic for future research. Supplemental Script S3 illustrates the calculation of IIF when several mechanisms co-exist. We tested whether a model assuming direct assortment as the sole source of partner similarity had worse fit to the data than a model with correlations estimated independently for each relationship type, with no assumptions on the source of similarity. We conducted a likelihood-ratio test with 1 degree of freedom. Among couples where both partners had a sibling, the sibling-in-law relations at each side of the family were modelled with the same correlation, whereas the co-sibling-in-law correlations were estimated freely. All models were adjusted for mean sex differences. We accounted for multiple testing and obtained False Discovery Rate (FDR) adjusted $p$-values using the Benjamini-Hochberg method with the $p.adjust()$ function in R.

The tetrachoric correlations rely on an underlying normal distribution. If the underlying distribution is left-skewed, the tetrachoric correlations can become overestimated. The product of correlations is pivotal for testing deviations from direct assortment. We therefore conducted simulations to determine whether left-skewness affected the product of the correlations (Supplemental Scripts S2, S3). This was not the case. Hence, if the Pearson correlations $r_{ab} * r_{bc} = r_{ac}$ for continuously non-normally distributed variables, then $r'_{ab} * r'_{bc} = r'_{ac}$ holds true for their dichotomized tetrachoric counterparts, even if the individual tetrachoric correlations are overestimated.

To obtain residual partner correlations after accounting for similarity in education (aim 4), we additionally adjusted the above models for either GPA or EA. This is equivalent to fitting a structural equation model (SEM) to the illustration in panel D of Fig. 1, where $A_1$ and $A_2$ represent the traits of interest and $B_1$ and $B_2$ represent the two partners' education. This does not impose any assumptions on why traits are correlated within an individual. All analyses were run with the

A) Direct assortment

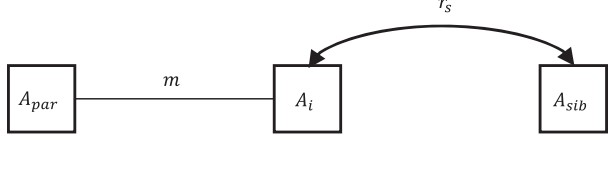

C) Social stratification

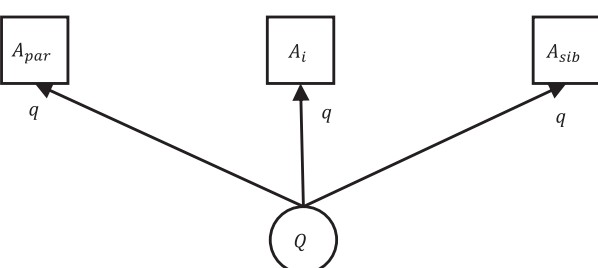

B) Indirect assortment based on unknown trait

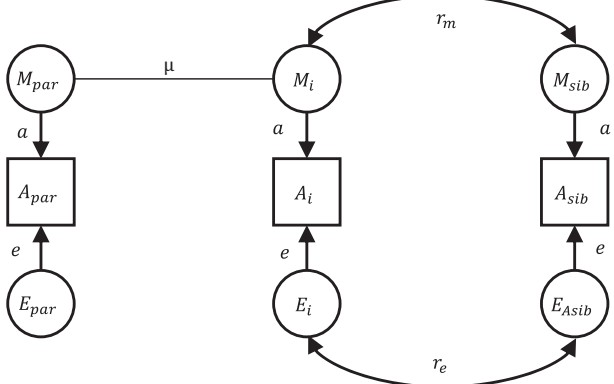

D) Convergence

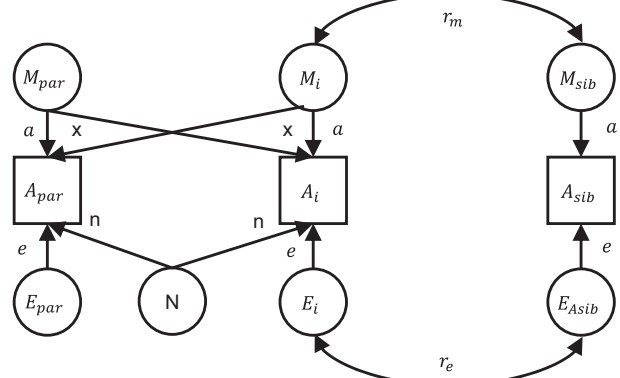

**Fig. 7 | Mechanisms of similarity between partners, siblings, and in-laws.** Four mechanisms of partner similarity and expected correlations between partners, siblings, and siblings-in-law. $A_i$ is the index person's phenotype, $A_{par}$ is their partner's phenotype, and $A_{sib}$ is their sibling's phenotype. **A** illustrates direct assortment on phenotype A. **B** illustrates indirect assortment, where partner similarities in A is due to assortment on the latent variable M and siblings may also correlate in residual variance (E). **C** illustrates social stratification (Q). **D** illustrates convergence by shared environments (N) or mutual influences (x). Unit variance can be assumed for simplicity.

health conditions measured prospectively 10 to 5 years before parenthood and again cross-sectionally with health observed in 2015–2019. Using the prospective data, we also estimated the associations between partners, siblings, and siblings-in-law as odds ratios using multiple logistic regression, adjusting for each individual's phenotype. The adjusted association with the siblings-in-law's phenotype tests direct assortment, reasoning that if assortment is based on the phenotype, then the siblings-in-law's phenotype should not relate to the index person's trait once we account for the partner's phenotype.

### Reporting summary
Further information on research design is available in the Nature Portfolio Reporting Summary linked to this article.

## Data availability
The data for this study encompass educational outcomes and primary care records for entire cohorts of the Norwegian population. The raw data are protected and are not available due to data privacy laws. Researchers can access the data by application to the Regional Committees for Medical and Health Research Ethics and the data owners (Statistics Norway and the Norwegian Directorate of Health). The authors cannot share these data with other researchers due to the sensitive nature and potential for identification. However, other researchers can contact the authors if they have questions concerning the data or overlapping research projects. Source data are provided with this paper.

## Code availability
The supplemental information contains scripts for estimating the in-law inflation factor under various forms of assortment (Script S1), and for testing potential bias in tetrachoric correlations for skewed variables (Script S2) and the product of skewed variables (Script S3).

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

## Acknowledgements

This work is part of the REMENTA and PARMENT projects and was supported by the Research Council of Norway (#300668 and #334093, respectively, to F.A.T.). The Research Council of Norway supported R.C., N.H.E., and E.Y. (#288083 and #336078, to E.Y.). This work was performed on the TSD (Tjeneste for Sensitive Data) facilities, owned by the University of Oslo, operated and developed by the TSD service group at the University of Oslo, IT-Department (USIT). This work was partly supported by the Research Council of Norway through its Centres of Excellence funding scheme, project number 262700. The project was co-funded by the European Union (ERC, BIOSFER, 101071773). Views and opinions expressed are however those of the author(s) only and do not necessarily reflect those of the European Union or the European Research Council. Neither the European Union nor the granting authority can be held responsible for them. This work was supported by the National Institute of Mental Health R01 Grants MH130448 and MH100141 (M.C.K.).

## Author contributions

F.A.T. conceived the idea and designed the models, with support from H.F.S., E.M.E., and M.C.K. F.A.T. carried out the analyses and visualised the results with support from H.F.S. R.C., N.H.E., M.C.K., and E.Y. contributed to interpretation of the results. F.A.T. wrote the manuscript with input from all authors. All authors provided critical feedback, discussed the results, helped shape the manuscript, and approved of the final manuscript.

## Funding

## Competing interests

The authors declare no competing interests.
