## [Transparent Peer Review file · Nature Communications]

Non-random mating patterns within and across education and mental and somatic health

Corresponding Author: Dr Fartein Torvik

Version 0:

Reviewer comments:

Reviewer #2

(Remarks to the Author)

Torvik et al present a thorough study on mental/somatic trait assortment in a specific Swedish population of couples. The manuscript is well-written and the presented analyses are rigorously conducted. The paper tackles questions of mental vs somatic similarity, cross-trait vs same-trait assortment strength, EA/GPA as key drives of indirect assortment, couple convergence. The in-law inflation is a smart approach to detect indirect assortment (it's a pity that these deviations are not used to correct observed correlations). The conclusions are mostly well-justified and the Discussion of the paper is detailed and nuanced. Below I list some major and minor points that could be addressed to further improve this work.

Major comments:

The test for direct assortment is neat, but it does not answer the questions where the indirect impact comes from. It can be an indirect path via other traits in the partners, or additional path between other family members involved. Most likely it is due to a confounder shared across the 3 persons (couple+sib), e.g. socio-economic factors. It would be nice if the authors could dig deeper on this, otherwise the indirect assortment remains vague. Can these in-law inflations be used to correct the couple correlation for potential confounders?

Another type of indirect effect that could be tested and the authors have already generated all the data for this is the cross-trait correlations more in general. E.g. if $\text{corr}(\text{traitA}(\text{couple person 1}), \text{traitB}(\text{couple person 2})) = \text{corr}(\text{traitA}(\text{couple person 1}), \text{traitB}(\text{couple person 1})) * \text{corr}(\text{traitB}(\text{couple person 1}), \text{traitB}(\text{couple person 2}))$ and $\text{corr}(\text{traitA}(\text{couple person 1}), \text{traitB}(\text{couple person 2})) = \text{corr}(\text{traitA}(\text{couple person 1}), \text{traitB}(\text{couple person 1})) * \text{corr}(\text{traitA}(\text{couple person 1}), \text{traitA}(\text{couple person 2}))$. Where these equalities cannot be rejected are uninteresting situations, because the "indirect assortment" mechanism is obvious (for the first one, it is driven by assortment on trait B, while for the second it is driven by assortment on trait A). I'd highly encourage the authors to explore this further to strengthen the message of the paper. This is kind of what they did but only explored whether EA/GPA can explain indirect assortments.

The adjustment for EA/GPA would only be correct if the authors could prove that these traits have causal effect on the other examined traits (and no reverse causal effect, nor confounding). This is most likely not the case, thus the correlation-based adjustment cannot necessarily be interpreted as evidence for indirect assortment via EA/GPA. The most obvious mistake could be to adjust for a collider (being EA). I don't think this weakness can be addressed, but at least must be admitted as a limitation.

"The correlations between partners' mental health conditions increased notably from a median of 0.14 in prospective analyses to a median correlation of 0.25 in cross-sectional analyses." – How can you be sure this is not the impact of having children? Also, can there be a selection bias that only those couples were considered who stayed together longer, a factor may be a correlate of couple similarity. Therefore, these results are not convincing and have to be toned down substantially.

Have the authors found any discrepancy between male trait A – female trait B and female trait A – male trait B correlations?

How were the 10 somatic traits selected? They seem rather arbitrary and classical diseases are not included (e.g. cardio-metabolic, anthropometric). Because of this very limited (and atypical) set of somatic variables selected require sentences

such as “Our study indicated that mental health conditions were more important than somatic health conditions for partner selection.” to be toned down.

Minor comments:

“However, final EA is often not obtained until after a couple meet. “ - This needs a reference as it can be highly generation and population specific.

“Population-wide data with no participation bias,” – Bias is expected to be much larger due to cultural differences or generational effects than participation. Can the authors comment on the magnitude of impact of participation bias on estimation assortative mating strength?

“5 to 10 years before a couple had their first child to minimize effects of convergence” – Still concretely, do you know how long these couples have been together? Can't you directly filter on that information rather than something indirect?

“For 20 of the 22 traits, partner correlations were statistically significant at the $\alpha=0.05$ level. “ – Multiple testing correction must be applied and anecdotal (nominal significance level) associations should be avoided (or max referred to in the supplement). This nominal level significance should be removed from everywhere in the manuscript.

“This was above 1.00 for 20 of 22 phenotypes, with statistically significantly deviations from direct assortment at the $\alpha=0.05$ level for GPA, EA, 3 mental health conditions, and 5 somatic health conditions” – Hypergeometric test should be done to convincingly show deviations to by chance event. (I'm convinced it will be significant, but such a test is much more convincing than reporting nominally significant deviations from 1, which tells nothing concrete.)

Based on what Table 3 is sorted? I was expecting to be sorted by the in-law inflation.

Figure 1 should have confidence intervals on the prevalences.

“Partner correlations in mental health were considerably higher at the end than at the start of the observational period.” – This requires a statistical testing with multiple testing adjustment to back it up. Also, this may be because the mental problems may be diagnosed later.

Reviewer #3

(Remarks to the Author)

The authors present an analysis of cross-trait similarity across mates and their relatives for a number of psychosocial, health, and disease phenotypes. Their results present additional evidence to several recent investigations showing that cross-mate assortative mating (AM) is ubiquitous--this is important for at least two reasons: 1. AM induces correlations between genetic liabilities beyond that due to pleiotropy / correlated effects, and 2. widely used polygenic estimators are misspecified and will over estimate both of these quantities. Beyond this, cross-trait AM also breaks Mendelian Randomization and impacts association statistics and polygenic score estimates. The manuscripts' greatest strength is that the authors were able to examine mate concordance around the time mates met, thus removing any effects of convergence (that said, the authors find limited evidence for substantial convergence, congruent with previous findings). All this said, I have some concerns, which I elaborate below.

Major concerns:

1. The concepts of primary vs secondary assortment are not well-defined in a multivariate context. The authors define these as:

“First, direct assortment (or primary phenotypic assortment) means that partners resemble each other in a trait because the observed trait influences partner selection. Direct assortment is a sufficient explanation for partner similarity in height. Second, indirect assortment (also called secondary assortment) refers to similarity in a trait resulting from selection on a correlated trait. This could be, for instance, psychiatric vulnerability for mental disorders or traits that are observed with measurement error”.

For two traits y_1, y_2 , denote the cross-mate cross-trait correlation matrix

$$W = ((r_{11}, r_{21}), (r_{12}, r_{22}))$$

where, e.g., r_{21} denotes the correlation between females' values on y_2 and males values on y_1 .

Only a fraction of the possible values of W can be represented by mating on a linear combination of y_1 and y_2 . When does W correspond to primary vs secondary mating in general? Would

$$W^* = ((r_{11}, 0), (0, r_{22}))$$

be direct assortment? Because W^* is only achievable by considering interactions between y_1, y_2 across mates.

Further, the role of measurement error isn't clear. Even in the univariate case, if the cross mate correlation is less than 1, individuals are mating on a linear combination of the phenotype and random noise. This is equivalent to mating on a trait measured with error. I think the authors would call this primary assortment despite this.

They later return to measure error, stating

"Measurement error is one source of indirect assortment, where individuals chose each other based on the true values of traits, which are imperfectly measured."

While you can define this mathematically, this concept is absurd and under this definition all mating should be indirect. How exactly do mates choose each other perfectly on the basis of true values? Because humans can perfectly measure each other on complex phenotypes like latent vulnerability to multiple psychiatric disorders? Again, if the correlation is less than 1 they are sorting on the trait and noise.

My recommendation would be that the authors distinguish between "complex mating" and the three edge cases of primary univariate phenotypic assortment, convergence, and social homogamy. Or alternatively, they can make this distinction between primary and secondary rigorous. As it stands, it's not at all clear what this distinction actually means in a multivariate context.

2. The role that transmission of phenotypes plays in the authors' analysis is unclear. For example, the authors state "assortment leads to correlations between all genetic and environmental influences in one partner and those in the other." This is only true when environment is transmitted (which of course it will usually be). But this will also affect cross-mate correlation structures, further complicating this whole direct vs indirect issue. As far as I can tell, the authors don't discuss this in depth, though it is certainly a limitation. Especially when talking about traits like educational attainment where parental factors (like income) will matter quite a bit.

Other concerns

3. I think the authors undersell the implications of their findings. For example, recent work has shown that beyond altering genetic correlations between true genetic liabilities, cross-trait AM biases GWAS test stats, inflates genetic correlation estimates (beyond real changes in architecture), and breaks MR (eg Brumpton et al. Nature Comm 2020, Border et al. Science 2022). This should be noted in the introduction and deserves more attention in the discussion. In particular, given the authors' findings that cross-trait correlation is ubiquitous across psychiatric disorders, this should complicate the interpretation of the 'p factor' literature.

4. The authors reportedly refer to the cluster of misfortunes, which isn't incorrect, but I think is easily misunderstood in light of previous (low-quality) arguments suggesting that AM will lead to a genetic underclass (in the bestselling P.O.S. The Bell Curve for example). It will also increase clustering of medium fortunes and good fortunes. Perhaps "social inequality" is a better term here than "clustering of disadvantages"?

5. (minor) Figure 1 is missing standard errors or some other visual quantification of uncertainty. Figure 3 is difficult to read as the text is inconsistently aligned. Also the majority of the color space from (-1,1) isn't represented on the plot so everything's pretty white and hard to distinguish.

Conclusions

Overall, the data presented by the author are extremely useful but I'm less convinced by the secondary analyses of these data and there are a couple of important topics missing from the discussion. Provided my concerns are addressed, this will make a valuable contribution to the literature.

Reviewer #4

(Remarks to the Author)

Torvik et al. Model causes of similarity among partners based on a large representative dataset. They leapfrog a lot of previous work in this area by looking at data in registers that describes a period that is plausibly before the partners ever met (which I'd call retrospective??). Their work is informative, is vital to the field and is worth reporting in a journal with broad reach. Precisely because the work has so much potential, I'll be highly critical of the flaws as I perceive them.

The results are interesting, but they at certain points fall short of convincing while they shouldn't have to fall short. I have a few suggestions that could significantly enhance the degree to which the statistical analysis of partner, sibling and in-law similarities inform the underlying hypothesized data generating mechanisms.

Let's begin by suggestions to clarify the relation between the mechanism and the statistical results. The mechanisms are currently listed in the intro, the results are presented in the results but without a way to link them to the mechanism, which is in the methods but then even sometimes hidden in references to previous work or supplemental figs/tables. What I am missing is a "key" or "roadmap" linking the proposed mechanisms to the statistics

Major Suggestion 1: write out a roadmap as a table or figure in the paper, now I am scrolling all over to figure out what a result means in terms of the model outlined in the intro and then to the methods to confirm I understood the relation between the statistic and the model.

Mechanisms discussed:

Social homogamy
Convergence
Primary assortment
Secondary assortment
Assortment conditional on assortment on EA/GPA

Statistics presented:

Cross sectional vs prospective (convergence pushes up cross sectional over prospective)
In-law inflation factor (goes up due to homogamy & indirect, goes down due to convergence)

Statistics in the supplement:

In-law inflation factor based on cross sectional data.

Issues I immediately notice:

The compute the in-law inflation factor on the prospective data, which BTW is a weird name for data collected deep in the past but that's an aside (maybe retrospective?) which means its very very unlikely to be pushed down due to convergence? If you compute the in-law inflation on contemporary data, then the change in in-law inflation might be convergence, under the authors model. So here is major recommendation two: display the inflation factor after convergence can have occurred in the main text table. Show that indeed in-law and sib correlations remain constantly correlated, but spouses go up as your convergence model implies. The pre/post comparison I now got from scrolling to Supplemental figure 3 is the core message: spouse converge, sibs and in-laws do not, this rules out any type of change in the confounding structure or C or G by life event interaction causing apparent convergence (because it would also lead to sib and in-law convergence). The reduction in the inflation is the test, which you can compute over groups of traits for power if you like.

Causal inference issue:

First you move to prospective analysis to control for convergence (which is great!), then you control for age (again good choice), finally you "control" the pairwise assortment computations for EA and GPA. You consistently retain some within and across mental trait.

Minor (but important) issue: Table 2 has no standard errors? That not really acceptable for your key result is it? Why doesn't the table split things out for the various the traits? In the discussion you remark that the physical traits are really just a sort of negative control, maybe at least split out the within trait (conditional) assortment for the MH traits in table 2?

which implies a causal relation, which I am sure exists, but it bears considering that within person mental health issues before graduation will likely influence GPA and EA.

In the conclusion you write:

1: "Although partner correlations could be partially explained as by-products of assortment related to education, this was not a primary explanation of partner correlations in mental health."

2: "Our prospective analyses and use of proper diagnoses indicate that there is assortment on the liability to mental disorders, as questioned by Yengo 11. The lack of correlations between partners' polygenic indices in previous studies is likely due to limited discovery samples and small effects of each causal variant, giving the polygenic indices low predictive value for mental health conditions. Our study indicated that mental health conditions were more important than somatic health conditions for partner selection. This is not surprising, given that mental health is linked with marriage and fertility 31 and could indicate desirability to potential partners."

I'd say especially the specific conclusion: "Our study indicated that mental health conditions were more important than somatic health conditions for partner selection" Implies that people select based on their mental health liabilities, which means you do not believe that the correlations you observe are only social homogamy, and that wouldn't imply any form of selection of a partner.

These are strong causal claims based on observational analyses, you don't present the causal diagram, and you do not go further than statistical control. There are options you have here like fixed effects (based on birthplace for example, or primary/high school) to control for shared social background, or you could sample pseudo partners that match the real partner in terms of age, GPA, EA, birthplace, parental income etc. etc. Even those enhanced controls would just be controls as far as I can tell, you could go even further and use instruments for EA/GAP (birth order?) but that's up to you.

Finally, your own paper established that there isn't just primary assortment on EA/GAP but more to it, social homogamy, or assortment on an unobserved trait. If assortment is on an unobserved trait, then EA/GPA are insufficient controls by definition, and an instrument won't fix that, if its social homogamy, then you are assuming the social homogamy is correlated

between traits (not entirely implausible) but would it have to be perfectly correlated?

Major recommendations that flow from this:

1. A causal diagram (DAG, other your choice)
2. Assumptions you make about confounding and measurement quality when you translate your statistics to conclusions according to your causal diagram, ideally in the results or intro not at the end in the limitations.
3. A clear description of the terminology used in the discussion, if you say “partner correlations in MH after accounting for EA/GPA” what do you mean you conclude either trait specific homogamy and/or assortment? If you say “partners selection” what do you mean? Only assortment but not homogamy? Selection implies choice and or individual action to the reader.

This paper is far more than simple cross section analysis, you have done outstanding work trying to learn about complex mechanisms with serious consequences for both biomedical research (psychiatric genetics) and the social sciences, I hope you can be equally rigorous about your causal model, your assumptions therein, and their limitations, if asked to review again I won't be satisfied by a simple summation of these limitations I think it's vital the reader understands the causal model you imply, what you can and cannot test in those models given the data, and at what points you need to make some leaps of faith (i.e. assumptions).

Minor:

Line 175: “contract” is supposed to be contrast, I think?

Version 1:

Reviewer comments:

Reviewer #2

(Remarks to the Author)

I congratulate the authors for the very thorough revision. The paper has strengthened a lot and I'm happy how my comments were addressed.

One minor point is that I mistakenly mentioned hypergeometric test, but I meant binomial test to see whether the number of nominally significant deviations from 1 are significant. This is the following logic: if one observes, say, 15 out of 22 tests with P-value < 0.05 , the probability to observe 15 or more such tests can be computed as `pbinom(15,22,.05,lower.tail = FALSE)` [R command]. This gives an idea of overall significance of the findings. But given that the authors have applied FDR correction, it is also fine.

I'm not sure what test the authors used to compare the medians of two sets of correlations. [“The correlations between partners' mental health conditions increased notably from a median of 0.14 in prospective analyses to a median correlation of 0.25 in cross-sectional analyses ($\Delta -2LL = 211.40$, $\Delta df = 10$, $p < 1.00e-99$).”] It looks like likelihood ratio test, but which are the two models (then it should be “ $-2 \times \Delta LL$ ”)?

Reviewer #3

(Remarks to the Author)

I am satisfied with the authors' revisions. The manuscript is much stronger and will make a valuable contribution to the current literature.

Reviewer #5

(Remarks to the Author)

1. The authors have now included a roadmap, presented in Figures 1 and 7, which effectively outlines all the potential mechanisms of partner assortment. This addition is very helpful. We have a few minor suggestions regarding these figures and the accompanying text: 1) In text of Fig 1E add 'N' as follows: “Furthermore, partners could share environments and experiences (N)”;

2) The authors may consider changing the order of the Figures A-D in Fig 7 to be consistent with the order of the those in Fig 1, i.e. Fig 7A: direct assortment, Fig 7B: indirect assortment, Fig 7C: social stratification, Fig 7D: convergence;

3) Should 'E' in Fig 7A actually be 'U' as in Fig 1B? Please also define in Fig 7 legend.;

4) The description of Fig 7 (line 522-556) mentions 'rs' a few times wrt Fig 7C and 7D, but this is called 'rm' in the Figure rather than 'rs', please correct and make consistent.

2. In terms of the potential impact of convergence, the cross-sectional partner correlations (2015–2019) are higher than the prospective unadjusted partner correlations. As a result of convergence, participants and partners are becoming more similar over time, which increases the partner correlation and reduces the in-law inflation factor (IIF). However, when compared with the IIF values in Supplementary Table 3, 6 out of the 22 IIFs for the cross-sectional measurements are higher than those for the prospective measurements. How do the authors explain this? Furthermore, if data on the duration of partner cohabitation is available the authors may consider adjusting for the effects of convergence?

Rujia Wang & Harold Snieder

Reviewer #6

(Remarks to the Author)

Reviewer #2 (Remarks to the Author):

Torvik et al present a thorough study on mental/somatic trait assortment in a specific Swedish population of couples. The manuscript is well-written and the presented analyses are rigorously conducted. The paper tackles questions of mental vs somatic similarity, cross-trait vs same-trait assortment strength, EA/GPA as key drives of indirect assortment, couple convergence. The in-law inflation is a smart approach to detect indirect assortment (it's a pity that these deviations are not used to correct observed correlations). The conclusions are mostly well-justified and the Discussion of the paper is detailed and nuanced. Below I list some major and minor points that could be addressed to further improve this work.

We thank the reviewer for a thorough reading of our manuscript, an overall positive evaluation, and for helpful suggestions. We address each of the comments below.

Major comments:

The test for direct assortment is neat, but it does not answer the questions where the indirect impact comes from. It can be an indirect path via other traits in the partners, or additional path between other family members involved. Most likely it is due to a confounder shared across the 3 persons (couple+sib), e.g. socio-economic factors. It would be nice if the authors could dig deeper on this, otherwise the indirect assortment remains vague. Can these in-law inflations be used to correct the couple correlation for potential confounders?

We have made the following changes to the paper to clarify possible sources of indirect assortment and the contributions of the current findings.

- 1) In the introduction, we now clearly present different mechanisms that can lead to partner similarity, with diagrams, and explicitly state how each mechanism is handled in our study (see Figure 1 and line 78-95).

Figure 1:

B) Indirect assortment based on unknown trait

Indirect assortment, also referred to as secondary assortment, refers to the situation that partners resemble each other in a focal trait (A) due to assortment in correlated traits. The identity of the correlated traits may or may not be known to the researcher. When it is not known, it may be treated as a latent variable, and its role be deduced from correlational systems. Assortment may take place directly on an unidentified or composite phenotype M_A , but the researcher only observes A. When the assorted phenotype is unknown, variance in a trait can be separated into mated (M_A) and non-mated (U) variance, and the association between M_A and A can be estimated. This requires additional information, such as data on siblings-in-law or co-siblings-in-law. If measurement error is not taken care of, assortment on a phenotype measured with error is indistinguishable from indirect assortment based on an unknown trait. For example, assortment may be direct for depression, but indirect for a low-quality screening instrument. In this paper, indirect assortment is one of two possible explanations of deviations from direct assortment.

C) Indirect assortment based on known trait

With information on traits correlated with the focal trait, the degree of indirect assortment can be explicitly modelled. In the depiction, assortment on trait A is thought to be secondary to assortment on trait B. The assignment of traits as A and B depends on theory, as the model cannot statistically distinguish between them. For example, partners may be similar in their level of depression due to assortment on neuroticism. The residual mating (m_E) is an upper bound estimate of direct assortment because indirect assortment may also take place on additional traits. In this paper, we test to what degree similarity in health can be secondary to assortment on education.

D) Social stratification

Social stratification (Q) refers to partner selection taking place within constrained sections of the population. For example, regions could have varying levels of educational attainment, without the education itself influencing partner choice. This is the non-genetic equivalent to population stratification and is one of the meanings of the term “social homogamy” in the literature. In this paper, social stratification is one of two possible explanations of deviations from direct assortment.

Figure 1. Conceptual representations of mechanisms of similarity between partner 1 and 2 in trait
A. Several mechanisms could co-exist, and the list does not include complex multivariate assortment.

Line 78-95:

Partner similarity can arise from several potentially co-occurring processes. In Figure 1, we outline these processes and the role they play in the present paper. First, *direct assortment* (or *primary phenotypic assortment*) means that partners resemble each other in a trait because the observed trait influences partner selection (panel A). Direct assortment is a sufficient explanation for partner similarity in height²⁰⁻²². Second, *indirect assortment* (also called *secondary assortment*) refers to similarity in a trait resulting from selection on a correlated trait, which may be unknown (panel B) or known (panel C). For instance, similarity in a specific mental disorder could arise from assortment on psychiatric vulnerability. If one trait, such as attractiveness, underlies assortment for multiple other traits, cross-trait assortment can be observed for these other traits. Direct assortment on an imperfectly measured phenotype can statistically resemble indirect assortment on an unobserved phenotype²⁰. In such cases, assortment may be said to be direct for the trait of interest, but indirect for the indicator. Third, *social stratification* (or *social homogamy*) refers to individuals selecting each other based on environmental proximity, which incidentally make them similar in the phenotype of interest (panel D). Social stratification has been found to play a small to moderate role in partner similarity in EA²³⁻²⁵. Fourth, *convergence* refers to partners becoming more similar over time, either because they influence each other or because they share environments (panel E). Convergence has been found for lifestyle choices such as alcohol consumption and exercise²⁶. Convergence is not a form of assortment, but an alternative explanation of partner similarity.

- 2) In the methods, we present models of extended family similarity under the different mechanisms of partner similarity (see Figure 7 and line 522-556).

Figure 7:

Figure 7. Four mechanisms of partner similarity and expected correlations between partners, siblings, and siblings-in-law. A_i is the index person's phenotype, A_{par} is their partner's phenotype, and A_{sib} is their sibling's phenotype. Panel A illustrates direct assortment. Panel B illustrates social stratification. Panel C illustrates indirect assortment. Panel D illustrates convergence by shared environments (N) or mutual influences (x). Unit variance can be assumed for simplicity.

Line 522-556:

By testing if the observed correlations adhered to this pattern, we can detect deviations from direct assortment. We define an in-law inflation factor (IIF) as the ratio of the correlation between siblings-in-law to the expectation under direct assortment

$$IIF = \frac{r_{inlaw}}{r_{sibling} * r_{partner}}$$

Figure 7 displays four processes that can lead to partner similarity and allows path tracing expected correlations, and hence IIF, in the different scenarios. We here consider their implications in isolation, although combinations of the mechanisms are possible. Direct assortment (panel A of Figure 7) assumes that partnerships are based on the observed phenotype. By applying path tracing rules

allowing for co-paths³⁶ to Figure 7, one can see that $r_{partner} = m$, $r_{sibling} = r_s$, and $r_{inlaw} = mr_s$.

Hence, $IIF = \frac{r_{inlaw}}{r_{partner} * r_{sibling}} = \frac{mr_s}{m * r_s} = 1$, and deviations from 1.00 are not consistent with direct

assortment. For simplicity, we assume unit variance. Social stratification (panel B of Figure 7)

influences all individuals to the same degree, and in isolation leads to $r_{partner} = q^2$, $r_{sibling} = q^2$,

$r_{inlaw} = q^2$, and $IIF = \frac{r_{inlaw}}{r_{partner} * r_{sibling}} = \frac{q^2}{q^2 * q^2} = \frac{1}{q^2}$. Hence, $q \neq 0$ leads to an IIF above 1.00 ($|q| <$

1). With indirect assortment (panel C of Figure 7), partners are similar due to assortment based on

an unknown (latent) phenotype. Siblings are similar in the same latent phenotype (r_s) and may share

additional similarity ($E = (1 - a^2)r_e$). In isolation, indirect assortment gives $r_{partner} = a^2\mu$,

$r_{sibling} = a^2r_s + E$, $r_{inlaw} = a^2\mu r_m$, and $IIF = \frac{r_{inlaw}}{r_{partner} * r_{sibling}} = \frac{a^2\mu r_m}{a^2\mu * (a^2r_m + E)} = \frac{1}{a^2 + \frac{E}{r_m}}$. When there

is not a residual correlation between siblings ($r_e=0$), this reduces to $IIF = \frac{1}{a^2}$, which is always ≥ 1.00

as long as $-1 < a < 1$. A large value for E reduces IIF, which will be 1.00 if $r_e=r_m$, and possibly

below 1.00 if $r_e > r_m$. However, we consider large values for E to be rare, because it is residualized on

the component of a trait that matters to other individuals through mate selection and is further

reduced by measurement error. Convergence (panel D of Figure 7) can refer to two processes

increasing partner similarity. Shared environments exclusively influence partner correlations (n^2),

whereas mutual influences primarily influence partner correlations ($2xa$) while having a smaller

effect on in-law correlations ($xr_s a$). Because convergence primarily influence partner correlations, it

will increase the denominator and reduce IIF. We do not consider convergence here further, as we

deal with by design.

In sum, an IIF above 1.00 can be explained by both social stratification and indirect assortment,

although we cannot distinguish between these processes without additional data. This is a topic for

future research. Supplemental Script S3 illustrates the calculation of IIF when several mechanisms co-exist.

- 3) The current analyses of Norwegian data do not provide a final answer to why there are deviations from direct assortment, however, we are able to narrow the options. Considering that convergence is an implausible explanation for partner similarity in the prospective analyses, we are left with two mechanisms that can explain deviations from direct assortment: a) indirect assortment resulting from (direct) assortment on correlated phenotypes or b) social stratification/homogamy. It is not straightforward to separate between these, in part because we expect them to lead to very similar in-law correlations. In principle, it could perhaps be done with data from additional relatives, but that would change

the sample and considerably complicate the modelling. Like the reviewer, we believe that identifying the underlying mechanism is an important research question. We have updated the discussion on indirect assortment versus measurement error versus social stratification (homogamy) on line 324-354.

Although the phenotypic model could be falsified, the underlying mechanisms remain elusive. Both indirect assortment and social stratification³⁸ could increase in-law correlations disproportionately and explain our observations. In any case, partner resemblance is not solely due to assortment based on the observed phenotypes. Whether parts of the partner correlations in mental health are due to causal influences on partner choice remains to be determined. Identifying the traits that actively determine assortment is an important question for future studies. It might be more strongly related to general vulnerability to psychopathology³¹ than to specific disorders. Due to the strong cross-trait assortment, such causal effects may be more plausible at the level of general mental health, rather than for specific diagnoses. A previous study indicated that partner similarity in many traits was driven by assortment on a few key traits³⁸, but it did not include mental disorders. Future studies may explore whether partner resemblance across many traits can be more parsimoniously explained by assortment on one or a small number of dimensions.

Indirect assortment need not be based on symmetric assortment on a manifest phenotype. Measurement error can be indistinguishable from indirect assortment on an unknown trait. Assortment may then be said to be direct for the true values of a trait, but indirect for an imperfect indicator. As measurement error is widespread and relatively easy to estimate, accounting for measurement error could improve future studies on assortment. Indirect assortment could also be related to impression management, whereby partner selection could take place on successful misrepresentations of one's characteristics. This should, however, not influence sibling correlations. Finally, correlations in trait preferences among siblings can increase correlations between distant affines, such as co-siblings-in-law³². Hence, models of preferences may be needed to fully understand similarities in wider family networks.

Assortment leads to correlations between all genetic and environmental influences in one partner and those in the other. When parental traits leave a mark on their children through vertical transmission, this assortment leads to an intertwining of genetics and environment in the children. This can substantially increase gene-environment correlations in the child generation, which again increases the genetic similarity between partners⁹ (formula S1.8). If there is indirect assortment, the

partner similarity in assorted factors will be larger than indicated by the observed variables, and the intergenerational consequences can be underestimated.

- 4) In the discussion, we call for future studies on the causality of partner choice, on line 326-329.

In any case, partner resemblance is not solely due to assortment based on the observed phenotypes. Whether parts of the partner correlations in mental health are due to causal influences on partner choice remains to be determined. Identifying the traits that actively determine assortment is an important question for future studies.

Another type of indirect effect that could be tested and the authors have already generated all the data for this is the cross-trait correlations more in general. E.g. if $\text{corr}(\text{traitA}(\text{couple person 1}), \text{traitB}(\text{couple person 2})) = \text{corr}(\text{traitA}(\text{couple person 1}), \text{traitB}(\text{couple person 1})) * \text{corr}(\text{traitB}(\text{couple person 1}), \text{traitB}(\text{couple person 2}))$ and $\text{corr}(\text{traitA}(\text{couple person 1}), \text{traitB}(\text{couple person 2})) = \text{corr}(\text{traitA}(\text{couple person 1}), \text{traitB}(\text{couple person 1})) * \text{corr}(\text{traitA}(\text{couple person 1}), \text{traitA}(\text{couple person 2}))$. Where these equalities cannot be rejected are uninteresting situations, because the “indirect assortment” mechanism is obvious (for the first one, it is driven by assortment on trait B, while for the second it is driven by assortment on trait A). I’d highly encourage the authors to explore this further to strengthen the message of the paper. This is kind of what they did but only explored whether EA/GPA can explain indirect assortments.

Our updated approach accommodates this comment. Please see Figure 1 panel C for an illustration of bivariate assortment. The reviewer’s comment is nested in this approach, as the reviewer points to the situation where $m_{\epsilon}=0$, and where partner similarity in a trait entirely driven by assortment on another trait. We estimate both assortment on the background factor (education) and the residual partner similarity after adjustment.

This resembles what we already did, but we now explicitly use this approach. It is true that we only explored whether EA or GPA could explain indirect assortment for other traits. This is an advancement over previous studies, and we do not have other plausibly better “candidate traits” in our data, as these should in general have higher partner correlations than the traits subject to secondary assortment. It is computationally infeasible to test if all 20 dichotomous traits explain partner similarity in all other 20 dichotomous traits, as this would lead to sparse data.

Please see Figure 1 panel C and the methods, line 573-577.

To obtain residual partner correlations after accounting for similarity in education (aim 4), we additionally adjusted the above models for either GPA or EA. This is equivalent to fitting a structural equation model (SEM) to the illustration in panel D of Figure 1, where A_1 and A_2 represent the traits of interest and B_1 and B_2 represent the two partners' education. This does not impose any assumptions on why traits are correlated within an individual.

The adjustment for EA/GPA would only be correct if the authors could prove that these traits have causal effect on the other examined traits (and no reverse causal effect, nor confounding). This is most likely not the case, thus the correlation-based adjustment cannot necessarily be interpreted as evidence for indirect assortment via EA/GPA. The most obvious mistake could be to adjust for a collider (being EA). I don't think this weakness can be addressed, but at least must be admitted as a limitation.

We do not make claims of within-individual influences between traits. As can be seen from Figure 1 panel C, the expected correlation matrix does not depend on the causal structure between trait A and B. If the two-arrowed paths were replaced with influenced from B to A, the expected co-variance matrix would not change.

The reviewer is right that we cannot conclude that EA or GPA determine partner selection, as we did not do a causal analysis. There could be other traits that determines partner selection and that leads to similarity in EA/GPA as well as the health traits. Likely it is, as we find indirect assortment also for EA and GPA. Still, it is interesting to test to what degree assortment is secondary to EA/GPA because even if we cannot prove causality, we demonstrate that assortment on these variables is related. If there was no residual partner similarity in e.g., mental health after adjustment for EA, researchers could focus simply on EA and drop assortment on mental health as a topic.

We have updated the discussion on line 383-391:

The current study indicates, as do also previous studies^{20,25}, that the strong partner resemblance in EA is due to even stronger resemblance in an unobserved factor. This was also the case for GPA. Assortment for EA and GPA was itself indirect; therefore, unidentified factors must exist that

contribute to partner similarity in education as well as in health. This aligns with a previous study that chain-linked in-laws and inferred far greater partner similarity in latent (unidentified) advantages than in the observed level of education²⁵. Our inclusion of GPA early in life is novel, however, it did not capture these latent factors any better than EA. Future research could try to identify traits that account for the sorting process and understand how they relate to partner similarity across observed traits. This may include social status more broadly defined or health in childhood.

“The correlations between partners’ mental health conditions increased notably from a median of 0.14 in prospective analyses to a median correlation of 0.25 in cross-sectional analyses.” – How can you be sure this is not the impact of having children? Also, can there be a selection bias that only those couples were considered who stayed together longer, a factor may be a correlate of couple similarity. Therefore, these results are not convincing and have to be toned down substantially.

Convergence refers to the tendency of partners to become similar over time. Having children is one possible source of convergence, as an example of shared experience/environment. We acknowledge that Supplemental Figure S12 of the initial submission only included effects of one partner on the other as the illustration of convergence. However, with the revised Figure 1, it should now be clear that shared environments and experiences, such as having a child, is also a source of convergence.

We have also updated the introduction on line 92-93:

Fourth, *convergence* refers to partners becoming more similar over time, either because they influence each other or because they share environments (panel E).

Have the authors found any discrepancy between male trait A – female trait B and female trait A – male trait B correlations?

The full correlation matrix is available in Figure 4 and summarized in Table 2, including sex differences. We now note in the results on line 194-195:

We did not observe any noteworthy differences in the correlations between male and female traits.

How were the 10 somatic traits selected? They seem rather arbitrary and classical diseases are not included (e.g. cardio-metabolic, anthropometric). Because of this very limited (and atypical) set of

somatic variables selected require sentences such as “Our study indicated that mental health conditions were more important than somatic health conditions for partner selection.” to be toned down.

The traits were selected based on a) presence in the ICPC-2 manual; b) prevalence in the sample, that is, among young adults seeking medical care for the condition 5-10 years before they have their first child; d) distinctiveness, that is, covering different aspects of somatic health. Anthropometric conditions were not included because there were no good diagnostic indicators in the ICPC-2 manual. Cardiometabolic conditions (and malignant tumors) had low prevalence among young adults to warrant further study. Due to the overall good somatic health in young adulthood, and perhaps also some selection of healthy individuals into parenthood, the traits mentioned by the reviewer could not be included.

We agree that the cited sentence was too general and have changed it to the following (line 269-270):

Our study indicated that mental health conditions were more strongly related to partner selection than somatic health conditions common in young adulthood.

We have updated the discussion on line 399-404:

We could only study somatic conditions that were common among parents-to-be in young adulthood. The results are not necessarily representative for other somatic health conditions; in particular, assortment on rare health conditions is unknown. This also prevented the study of health conditions with a higher average age of onset, such as cardiometabolic conditions and cancers. However, conditions that develop after couple formation cannot directly influence its composition.

We have updated the methods on line 498-500:

They were selected for their diversity in covering different health issues and for being sufficiently prevalent in both sexes in our sample of young adults who later became parents.

Minor comments:

“However, final EA is often not obtained until after a couple meet. “ - This needs a reference as it can be highly generation and population specific.

This was worded too strongly and did not communicate our intention, which was to ensure a couple is unlikely to influence each other in the variables that we study. This is much less likely to be the case for GPA obtained at age 16 than for educational attainment, which is necessarily obtained at a higher age for anyone going beyond lower secondary school, and sometimes at high ages. We have revised this sentence on line 110-111.

However, high EA is achieved in adulthood, potentially after meeting a partner, and may be influenced by convergence.

“Population-wide data with no participation bias,” – Bias is expected to be much larger due to cultural differences or generational effects than participation. Can the authors comment on the magnitude of impact of participation bias on estimation assortative mating strength?

We agree that strengths of assortment could vary between cultures. Generalizability is discussed as the second limitation of the study, line 404-406.

Second, our focus on parents of children born in Norway between 2016 and 2020 could limit the generalizability to other populations or time periods.

When it comes to participation bias, one may expect selective participation to downwardly bias the correlation between partners, because there will be fewer observed complete cases in the lower end of the distribution and hence a restriction of range. As both educational attainment and health predicts participation in cohort studies, this could be relevant for our results. We have updated the discussion on page 264-269.

An alternative explanation is that overrepresentation of healthy and well-educated individuals in cohort studies restricts the range and downwardly bias partner correlations. For example, we observed a partner correlation of 0.48 for EA, compared to 0.42 in a Norwegian cohort²⁰. However, for mental health, our estimates of correlations between partners-to-be was slightly lower (median $r=0.13$) than in a cohort study assessing global mental health among future partners ($r=0.16$)²⁹.

“5 to 10 years before a couple had their first child to minimize effects of convergence” – Still concretely, do you know how long these couples have been together? Can’t you directly filter on that information rather than something indirect?

Unfortunately, we do not know how long couples have been together. The register data only provides information on marriage or birth of a child. Likely, many couples were together for some time before they married or had their first child. We therefore observe parents-to-be many years before they have their first child, assuming that the level of convergence was minimal at that time. This is acknowledged as the third limitation of the study (line 406-411).

Third, we cannot rule out that some partners had already influenced each other at the start of the observational period in early adulthood. Nevertheless, the prospective nature of our study is a major advancement over previous studies, and the comparison with cross-sectional data emphasizes the impact of this analytic decision. The gap of 5 years between the end of health observation and the birth of the first child exceeds the median duration of relationships, suggesting that most couples were unacquainted during the health observation period.

We have also added the age at the start of follow-up to the results section, on line 138-139.

Women were on average 19.61 and men 21.96 years old at the start of the observational period.

“For 20 of the 22 traits, partner correlations were statistically significant at the $\alpha=0.05$ level. “ – Multiple testing correction must be applied and anecdotal (nominal significance level) associations should be avoided (or max referred to in the supplement). This nominal level significance should be removed from everywhere in the manuscript.

We now apply False Discovery Rate (FDR) adjusted p-values throughout the manuscript and have removed unnecessary references to statistical significance. We have also updated all correlation matrices to reflect whether the correlations are statistically significant with FDR adjustment.

“This was above 1.00 for 20 of 22 phenotypes, with statistically significantly deviations from direct assortment at the $\alpha=0.05$ level for GPA, EA, 3 mental health conditions, and 5 somatic health conditions” – Hypergeometric test should be done to convincingly show deviations to by chance

event. (I'm convinced it will be significant, but such a test is much more convincing than reporting nominally significant deviations from 1, which tells nothing concrete.)

The reviewer describes a scenario of multiple hypothesis tests being conducted. Hypergeometric tests are not typically used to account for multiple testing, so without further details it is not clear to us how they apply to this scenario. One hypergeometric test per trait would leave the multiple testing issue. We have applied FDR adjustment of p-values to the test of indirect assortment and updated the text accordingly.

Line 218-220:

False discovery rate adjusted p-values indicated statistically significant deviations from direct assortment at the $\alpha=0.05$ level for GPA, EA, 3 mental health conditions, and 4 somatic health conditions.

Line 561-563:

We accounted for multiple testing and obtained False Discovery Rate (FDR) adjusted p-values using the Benjamini-Hochberg method with the *p.adjust()* function in R.

Based on what Table 3 is sorted? I was expecting to be sorted by the in-law inflation.

All figures and tables have been re-drawn and the order of the variables are not consistent across all display items. The order is based on the within-trait partner correlations in prospective analyses, except that EA and GPA are shown first, as in Figure 3. This gives a consistent ordering of traits throughout the paper.

Figure 1 should have confidence intervals on the prevalences.

We have added confidence intervals to the figure, which is now Figure 2.

“Partner correlations in mental health were considerably higher at the end than at the start of the observational period.” – This requires a statistical testing with multiple testing adjustment to back it up. Also, this may be because the metal problems may be diagnosed later.

We have added significance tests to back this up on line 166-169.

The correlations between partners' mental health conditions increased notably from a median of 0.14 in prospective analyses to a median correlation of 0.25 in cross-sectional analyses ($\Delta-2LL = 211.40, \Delta df=10, p < 1.00e-99$). For somatic health conditions, the increases were more modest, from 0.04 to 0.06 ($\Delta-2LL = 63.05, \Delta df = 10, p = 9.55e-10$).

The word "considerably" is based on a subjective evaluation of the magnitude of these statistically significant differences. Regarding the interpretation, we assume the reviewer refers to mental problems being diagnosed later than their true onset. We have updated the limitations:

Line 394-399:

First, the medical records are proxies for actual health conditions, as not all individuals with health issues seek medical care. This prevented the study of conditions below the threshold of medical attention. This issue is reduced as the tetrachoric correlations model these thresholds. Also, our use of primary care data captures a larger proportion of cases than specialist care data alone³⁹, which has been used in previous studies². This further mitigates potential biases.

Reviewer #3 (Remarks to the Author):

The authors present an analysis of cross-trait similarity across mates and their relatives for a number of psychosocial, health, and disease phenotypes. Their results present additional evidence to several recent investigations showing that cross-mate assortative mating (AM) is ubiquitous--this is important for at least two reasons: 1. AM induces correlations between genetic liabilities beyond that due to pleiotropy / correlated effects, and 2. widely used polygenic estimators are misspecified and will over estimate both of these quantities. Beyond this, cross-trait AM also breaks Mendelian Randomization and impacts association statistics and polygenic score estimates. The manuscripts' greatest strength is that the authors were able to examine mate concordance around the time mates met, thus removing any effects of convergence (that said, the authors find limited evidence for substantial convergence, congruent with previous findings). All this said, I have some concerns, which I elaborate below.

We thank the reviewer for a thorough reading and thoughtful comments.

Major concerns:

1. The concepts of primary vs secondary assortment are not well-defined in a multivariate context.

The authors define these as:

"First, direct assortment (or primary phenotypic assortment) means that partners resemble each other in a trait because the observed trait influences partner selection. Direct assortment is a sufficient explanation for partner similarity in height. Second, indirect assortment (also called secondary assortment) refers to similarity in a trait resulting from selection on a correlated trait. This could be, for instance, psychiatric vulnerability for mental disorders or traits that are observed with measurement error".

For two traits y_1, y_2 , denote the cross-mate cross-trait correlation matrix

$$W = ((r_{11}, r_{21}), (r_{12}, r_{22}))$$

where, e.g., r_{21} denotes the correlation between females' values on y_2 and males values on y_1 .

Only a fraction of the possible values of W can be represented by mating on a linear combination of y_1 and y_2 . When does W correspond to primary vs secondary mating in general? Would

$$W^* = ((r_{11}, 0), (0, r_{22}))$$

be direct assortment? Because W^* is only achievable by considering interactions between y_1, y_2 across mates.

Further, the role of measurement error isn't clear. Even in the univariate case, if the cross mate correlation is less than 1, individuals are mating on a linear combination of the phenotype and random noise. This is equivalent to mating on a trait measured with error. I think the authors would call this primary assortment despite this.

They later return to measure error, stating

"Measurement error is one source of indirect assortment, where individuals chose each other based on the true values of traits, which are imperfectly measured."

While you can define this mathematically, this concept is absurd and under this definition all mating should be indirect. How exactly do mates choose each other perfectly on the basis of true values? Because humans can perfectly measure each other on complex phenotypes like latent vulnerability to multiple psychiatric disorders? Again, if the correlation is less than 1 they are sorting on the trait and noise.

My recommendation would be that the authors distinguish between "complex mating" and the

three edge cases of primary univariate phenotypic assortment, convergence, and social homogamy. Or alternatively, they can make this distinction between primary and secondary rigorous. As it stands, it's not at all clear what this distinction actually means in a multivariate context.

We are not aware of any authoritative source with good multivariate definitions of primary (direct) and secondary (indirect) assortment. We have added Figure 1 illustrate various forms of assortment. In Figure 1 panel C, we describe bivariate assortment, where there is indirect assortment for phenotype A (the focal phenotype) and direct assortment for another phenotype, B. We also allow for residual assortment on the parts of the focal phenotype (A) not related to B. This allows us to quantify the degree to which partner similarity in one variable can be explained by similarity in another variable. We believe this is an advancement over previous studies. Whereas mating on linear combinations of traits can take place, they are plausibly insufficient for explaining all cross-trait assortment, as variations in partner preferences are also important. Hence, we believe that a complete understanding of multivariate assortment would require theoretical advancements beyond the scope of this paper.

When it comes to treating measurement error as indirect assortment, it is commonly assumed in the literature that mating takes place on the observed phenotype, which includes measurement error. Hence, the mated and measured phenotypes could be different. For example, one may consider assortment to be direct for depression, but indirect for a poor depression index. It is not true that all assortment is indirect under this definition, because it would appear as direct once the responsible traits are found, and possibly adjusted for measurement error. Statistical measurement error is a different phenomenon than humans' evaluation of potential partners. It is possible that humans misjudge their partner's level of a certain trait – this would make the trait less important for partnership formation, but would not turn out as random measurement error, which would influence sibling and partners to the same degree.

We have made several changes throughout the manuscript to make these considerations clearer:

We have added Figure 1, which defines certain clear-cut mating mechanisms and included the following in its legend:

Several mechanisms could co-exist, and the list does not include complex multivariate assortment.

Introduction line 86-89:

Direct assortment on an imperfectly measured phenotype can statistically resemble indirect assortment on an unobserved phenotype ²⁰. In such cases, assortment may be said to be direct for the trait of interest, but indirect for the indicator.

Discussion line 306-307:

Modelling of variations in preferences may be vital to fully understand cross-trait assortment ^{3,32}.

Discussion line 337-346:

Indirect assortment need not be based on symmetric assortment on a manifest phenotype. Measurement error can be indistinguishable from indirect assortment on an unknown trait. Assortment may then be said to be direct for the true values of a trait, but indirect for an imperfect indicator. As measurement error is widespread and relatively easy to estimate, accounting for measurement error could improve future studies on assortment. Indirect assortment could also be related to impression management, whereby partner selection could take place on successful misrepresentations of one's characteristics. This should, however, not influence sibling correlations. Finally, correlations in trait preferences among siblings can increase correlations between distant affines, such as co-siblings-in-law ³². Hence, models of preferences may be needed to fully understand similarities in wider family networks.

2. The role that transmission of phenotypes plays in the authors' analysis is unclear. For example, the authors state

"assortment leads to correlations between all genetic and environmental influences in one partner and those in the other."

This is only true when environment is transmitted (which of course it will usually be). But this will also affect cross-mate correlation structures, further complicating this whole direct vs indirect issue. As far as I can tell, the authors don't discuss this in depth, though it is certainly a limitation. Especially when talking about traits like educational attainment where parental factors (like income) will matter quite a bit.

The cited sentence refers to the influences on the partners themselves. This can be seen in the top part of Figure 1 of Keller et al. (2009) [<https://www.cambridge.org/core/journals/twin-research-and-human-genetics/article/modeling-extended-twin-family-data-i-description-of-the-cascade-model/CEF68B8065BD879D907E9C4C60CE0931>], depicting the Nuclear Twin Family Design, where one can, for example tracing the path $A \rightarrow P(\text{Fa}) - P(\text{Mo}) \leftarrow E$. The consequences of these correlations

will naturally depend on childbearing and the degree of vertical transmission. We have updated the discussion related to intergenerational transmission on line 348-356:

Assortment leads to correlations between all genetic and environmental influences in one partner and those in the other. When parental traits leave a mark on their children through vertical transmission, this assortment leads to an intertwining of genetics and environment in the children. This can substantially increase gene-environment correlations in the child generation, which again increases the genetic similarity between partners⁹ (formula S1.8). If there is indirect assortment, the partner similarity in assorted factors will be larger than indicated by the observed variables, and the intergenerational consequences can be underestimated. Intergenerational studies therefore need to carefully model indirect assortment. Regardless of mechanism and possible genetic consequences of assortative mating¹⁸, the potential social consequences of partnership composition could remain.

Other concerns

3. I think the authors undersell the implications of their findings. For example, recent work has shown that beyond altering genetic correlations between true genetic liabilities, cross-trait AM biases GWAS test stats, inflates genetic correlation estimates (beyond real changes in architecture), and breaks MR (eg Brumpton et al. Nature Comm 2020, Border et al. Science 2022). This should be noted in the introduction and deserves more attention in the discussion. In particular, given the authors' findings that cross-trait correlation is ubiquitous across psychiatric disorders, this should complicate the interpretation of the 'p factor' literature.

We thank the reviewer for this suggestion. We have updated the abstract on line 30-31:

This has implications for the distribution of risk factors among children, for genetic studies, and for studies of intergenerational transmission.

We have updated the introduction on line 69-73:

Assortment across traits can lead to correlations between genetic⁴ and environmental influences on different traits¹⁷, which in turn can contribute to comorbidity and familial clustering of multiple disorders¹⁸. Beyond genuine increases in correlations between genetic liabilities, cross-trait assortment can also violate assumptions and bias genome-wide association and Mendelian randomization studies¹⁹.

We have updated the discussion on line 293-300:

The positive manifold across mental health conditions in partners can in the next generation increase genetic correlations between traits; not because the same set of genes are associated with different traits, but because genetic liabilities to different traits co-occur in the same individuals⁴. This can contribute to the frequently observed “p-factor”³¹. In addition, cross-trait assortment can easily lead to bias in genetic studies, as unmeasured genetic variants can be related to measured variants as well as the outcome of interest. This can inflate estimates in genome-wide association studies and violate the exclusion criteria in Mendelian randomization studies¹⁹. In the presence of cross-trait assortment, the results of such studies should be interpreted with caution.

4. The authors reportedly refer to the cluster of misfortunes, which isn't incorrect, but I think is easily misunderstood in light of previous (low-quality) arguments suggesting that AM will lead to a genetic underclass (in the bestselling P.O.S. The Bell Curve for example). It will also increase clustering of medium fortunes and good fortunes. Perhaps "social inequality" is a better term here than "clustering of disadvantages"?

We changed this to “clustering of education and health”, on line 45.

This provides insight into the clustering of education and health within families.

We also updated the introduction on line 69-71:

Assortment across traits can lead to correlations between genetic⁴ and environmental influences on different traits¹⁷, which in turn can contribute to comorbidity and familial clustering of multiple disorders¹⁸.

5. (minor) Figure 1 is missing standard errors or some other visual quantification of uncertainty.

We have added confidence intervals to the figure (which is now Figure 2).

Figure 3 is difficult to read as the text is inconsistently aligned. Also the majority of the color space from (-1,1) isn't represented on the plot so everything's pretty white and hard to distinguish.

We have redrawn all figures and have addressed this issue. The palette has been changed. The limits of the color space are set to reflect the largest (absolute) numbers present in the results. Negligible

differences in correlations may be indistinguishable by color, however, all correlations are written, in black if statistically significant after adjusting for false discovery rate, otherwise in gray.

Conclusions

Overall, the data presented by the author are extremely useful but I'm less convinced by the secondary analyses of these data and their are a couple of important topics missing from the discussion. Provided my concerns are addressed, this will make a valuable contribution to the literature.

We thank the reviewer for valuable input, which we believe have improved the manuscript several places.

Reviewer #4 (Remarks to the Author):

Torvik et al. Model causes of similarity among partners based on a large representative dataset. They leapfrog a lot of previous work in this area by looking at data in registers that describes a period that is plausibly before the partners ever met (which I'd call retrospective??). Their work is informative, is vital to the field and is worth reporting in a journal with broad reach. Precisely because the work has so much potential, I'll be highly critical of the flaws as I perceive them.

Th results are interesting, but they at certain points fall short of convincing while they shouldn't have to fall short. I have a few suggestions that could significantly enhance the degree to which the statistical analysis of partner, sibling and in-law similarities inform the underlying hypothesized data generating mechanisms.

Let's begin by suggestions to clarify the relation between the mechanism and the statistical results. The mechanisms are currently listed I the intro, the results are presented in the results but ithout a way to link them to the mechanism, which is in the methods but then even sometimes hidden in references to previous work or supplemental figs/tables. What I am missing is a "key" or "routemap" linking the proposed mechanisms to the statistics

We thank the reviewer for the overall positive response and helpful suggestions. Our responses are provided below.

Major Suggestion 1: write out a roadmap as a table or figure in the paper, now I am scrolling all over to figure out what a result means in terms of the model outlines in the intro and the to the methods to confirm I understood the relation between the statistic and the model.

Mechanisms discussed:

Social homogamy

Convergence

Primary assortment

Secondary assortment

Assortment conditional on assortment on EA/GPA

Statistics presented:

Cross sectional vs prospective (convergence pushes up cross sectional over prospective)

In-law inflation factor (goes up due to homogamy & indirect, goes down due to convergence)

Statistics in the supplement:

In-law inflation factor based on cross sectional data.

We thank for these suggestions. We have made several changes to the manuscript to provide a roadmap and to ensure that all relevant information is readily available without having to go back and forth.

With regard to the roadmap: We hope this new information serves as a useful roadmap.

We have made the These changes were made in response to this comment:

- We have added Figure 1 to the introduction, which illustrate the different mechanisms for partner similarity and explain how they are relevant to the paper. (Please see manuscript or above.)
- We have added Figure 7 to the Methods, which explains how the different testable mechanisms influence correlations between partners, siblings, and siblings-in-law. This is a revised version of Supplemental Figure 12 in the initial submission. (Please see manuscript or above.)

- Figure 7 is accompanied with a new paragraph in the methods section detailing how the mechanisms influence the test for deviations from direct assortment. Parts of the information now in the methods consists of improved versions of material that was previously in the supplement. The reader thus does not have to go back and forth between manuscript and supplement. Methods, line 522-556:

By testing if the observed correlations adhered to this pattern, we can detect deviations from direct assortment. We define an in-law inflation factor (IIF) as the ratio of the correlation between siblings-in-law to the expectation under direct assortment

$$IIF = \frac{r_{inlaw}}{r_{sibling} * r_{partner}}$$

Figure 7 displays four processes that can lead to partner similarity and allows path tracing expected correlations, and hence IIF, in the different scenarios. We here consider their implications in isolation, although combinations of the mechanisms are possible. Direct assortment (panel A of Figure 7) assumes that partnerships are based on the observed phenotype. By applying path tracing rules allowing for co-paths³⁶ to Figure 7, one can see that $r_{partner} = m$, $r_{sibling} = r_s$, and $r_{inlaw} = mr_s$.

Hence, $IIF = \frac{r_{inlaw}}{r_{partner} * r_{sibling}} = \frac{mr_s}{m * r_s} = 1$, and deviations from 1.00 are not consistent with direct

assortment. For simplicity, we assume unit variance. Social stratification (panel B of Figure 7)

influences all individuals to the same degree, and in isolation leads to $r_{partner} = q^2$, $r_{sibling} = q^2$,

$r_{inlaw} = q^2$, and $IIF = \frac{r_{inlaw}}{r_{partner} * r_{sibling}} = \frac{q^2}{q^2 * q^2} = \frac{1}{q^2}$. Hence, $q \neq 0$ leads to an IIF above 1.00 ($|q| <$

1). With indirect assortment (panel C of Figure 7), partners are similar due to assortment based on an unknown (latent) phenotype. Siblings are similar in the same latent phenotype (r_s) and may share additional similarity ($E = (1 - a^2)r_e$). In isolation, indirect assortment gives $r_{partner} = a^2\mu$,

$r_{sibling} = a^2r_s + E$, $r_{inlaw} = a^2\mu r_m$, and $IIF = \frac{r_{inlaw}}{r_{partner} * r_{sibling}} = \frac{a^2\mu r_m}{a^2\mu * (a^2r_m + E)} = \frac{1}{a^2 + \frac{E}{r_m}}$. When there

is not a residual correlation between siblings ($r_e=0$), this reduces to $IIF = \frac{1}{a^2}$, which is always ≥ 1.00

as long as $-1 < a < 1$. A large value for E reduces IIF, which will be 1.00 if $r_e=r_m$, and possibly

below 1.00 if $r_e > r_m$. However, we consider large values for E to be rare, because it is residualized on

the component of a trait that matters to other individuals through mate selection and is further

reduced by measurement error. Convergence (panel D of Figure 7) can refer to two processes

increasing partner similarity. Shared environments exclusively influence partner correlations (n^2),

whereas mutual influences primarily influence partner correlations ($2xa$) while having a smaller

effect on in-law correlations ($xr_s a$). Because convergence primarily influence partner correlations, it will increase the denominator and reduce IIF. We do not consider convergence here further, as we deal with by design.

In sum, an IIF above 1.00 can be explained by both social stratification and indirect assortment, although we cannot distinguish between these processes without additional data. This is a topic for future research. Supplemental Script S3 illustrates the calculation of IIF when several mechanisms co-exist.

- We have also added Supplemental Script S1 describing how various mechanisms influence IIF.
- We have moved adjusted results (Figures 5-6; adjusting for GPA and EA) from the supplement to the main text. Now, it should be possible to follow all relevant results from reading the main text. The supplement no longer includes information necessary to understand the main message. It exclusively details information that could be relevant for particularly curious readers.
- We have added guiding text to the results section, repeating which aim/mechanism each result is meant to address and changed the subheading of the results section to point to the aim they address (instead of summarizing the results). For example on line 200

Testing if associations are in line with direct assortment (aim 3)

The IIF for the cross-sectional analyses are still in the supplement as their interpretation is unclear. This is unlike the prospective analyses in the main text, where convergence is excluded as an explanation. We have added the following text to the legend for Supplemental Table S3.

* IIF is expected to equal 1.00 under direct assortment. The cross-sectional partner correlations can be influenced by convergence, generally reducing the IIF. This may cancel out effects of indirect assortment and social stratification, which generally increases IIF, thereby limiting the interpretability of IIF among established couples.

Issues I immediately notice:

The compute the in-law inflation factor on the prospective data, which BTW is a weird name for data collected deep in the past but that's an aside (maybe retrospective?) which means its very

very unlikely to be pushed down due to convergence? If you compute the in-law inflation on contemporary data, then the change in in-law inflation might be convergence, under the authors model. So here is major recommendation two: display the inflation factor after convergence can have occurred in the main text table. Show that indeed in-law and sib correlations remain constantly correlated, but spouses go up as your convergence model implies. The pre/post comparison I now got from scrolling to Supplemental figure 3 is the core message: spouse converge, sibs and in-laws do not, this rules out any type of change in the confounding structure or C or G by life event interaction causing apparent convergence (because it would also lead to sib and in-law convergence). The reduction in the inflation is the test, which you can compute over groups of traits for power if you like.

We believe the degree of convergence can more directly be assessed by comparing partner correlations in the two observational periods, which is depicted in Figure 3. While it is true that convergence can reduce the in-law inflation factor, convergence effects may not only be limited to partners. It is also possible that siblings (or siblings-in-law) influence each other or increasingly share environments. Convergence could thus also influence siblings, and to a smaller degree siblings-in-law (for example, mutual influences between partners is expected to slightly increase sibling-in-law correlations). Hence, increasing sibling correlations would not undermine or support our findings. We therefore believe this is not central to answer any of the 4 aims of the study (1: estimate within trait assortment; 2: estimate across trait assortment; 3: determine if correlations are in line with direct assortment; 4: determine if education could underlie indirect assortment in health). We control for convergence to avoid having it influence the results. This being said, the reviewer raises an interesting follow-up question. Our data have aspects resembling both prospective and retrospective designs. We chose to call the longitudinal analyses “prospective” to highlight that we utilize the fact that participants were followed continuously over time and the data collected as the events unfolded.

Causal inference issue:

First you move to prospective analysis to control for convergence (which is great!), then you control for age (again good choice), finally you “control” the pairwise assortment computations for EA and GPA. You consistently retain some within and across mental trait.

which implies a causal relation, which I am sure exists, but it bears considering that within person mental health issues before graduation will likely influence GPA and EA.

We agree that mental health early in life could influence GPA and EA. If this happened for both partners, and if EA/GPA influence partner selection, then the contributions of mental health to partner selection can be underestimated (similarly to adjusting for a mediator). If this bias is present, the true associations between mental health in partners would be larger than indicated by our findings. It would therefore not change the conclusion that mental health is related to couple formation. This is more likely a threat for analyses of EA than of GPA. For EA, this could go on after couple formation, whereas GPA is determined at age 16, before couple formation (for the vast majority of couples). We have updated the discussion on line 372-375:

Using GPA as an alternative indicator of educational potential reduces this issue because it is typically achieved before partners meet. However, each partner's mental health could have influenced their own GPA, meaning that the true assortment on mental disorders could in principle be slightly larger than indicated by our study.

We have updated the discussion on line 384-389:

This was also the case for GPA. Assortment for EA and GPA was itself indirect; therefore, unidentified factors must exist that contribute to partner similarity in education as well as in health. This aligns with a previous study that chain-linked in-laws and inferred far greater partner similarity in latent (unidentified) advantages than in the observed level of education²⁵. Our inclusion of GPA early in life is novel, however, it did not capture these latent factors any better than EA.

Minor (but important) issue: Table 2 has no standard errors? That not really acceptable for your key result is it? Why doesn't the table split things out for the various the traits? In the discussion you remark that he physical traits are really just a sort of negative control, maybe at least split out the within trait (conditional) assortment for the MH traits in table 2?

We have updated the table with median standard errors for correlations within and across traits in different categories. The exact correlations for all combinations of the various traits are available in Figures 4, 5, and 6, and Supplemental Figures.

In the conclusion you write:

1: **“Although partner correlations could be partially explained as by-products of assortment related to education, this was not a primary explanation of partner correlations in mental health.”**

2: **“Our prospective analyses and use of proper diagnoses indicate that there is assortment on the liability to mental disorders, as questioned by Yengo 11. The lack of correlations between partners’ polygenic indices in previous studies is likely due to limited discovery samples and small effects of each causal variant, giving the polygenic indices low predictive value for mental health conditions. Our study indicated that mental health conditions were more important than somatic health conditions for partner selection. This is not surprising, given that mental health is linked with marriage and fertility 31 and could indicate desirability to potential partners.”**

I’d say especially the specific conclusion: **“Our study indicated that mental health conditions were more important than somatic health conditions for partner selection”** Implies that people select based on their mental health liabilities, which means you do not believe that the correlations you observe are only social homogamy, and that wouldn’t imply any form of selection of a partner.

These are strong causal claims based on observational analyses, you don’t present the causal diagram, and you do not go further than statistical control. There are options you have here like fixed effects (based on birthplace for example, or primary/high school) to control for shared social background, or you could sample pseudo partners that match the real partner in terms of age, GPA, EA, birthplace, parental income etc. etc. Even those enhanced controls would just be controls as far as I can tell, you could go even further and use instruments for EA/GAP (birth order?) but that’s up to you.

Finally, your own paper established that there isn’t just primary assortment on EA/GAP but more to it, social homogamy, or assortment on an unobserved trait. If assortment is on an unobserved trait, then EA/GPA are insufficient controls by definition, and an instrument won’t fix that, if its social homogamy, then you are assuming the social homogamy is correlated between traits 9not entirely implausible) but would it have to be perfectly correlated?

We thank the reviewer for this comment and for making us aware that we used causal language in some instances even though our study indicated indirect assortment. We have therefore 1) changed the language in the cited sections and elsewhere in the manuscript to avoid causal language [e.g., “more important” to “more strongly related to”]; 2) expanded the discussion with a section on causality and direct assortment (see this reviewer’s comment #2).

Major recommendations that flow from this:

1. A causal diagram (DAG, other your choice)

Please see Figure 1 of the introduction and Figure 7 of the methods section and the corresponding text on lines 78-95 and 522-556 (pasted above).

2. Assumptions you make about confounding and measurement quality when you translate your statistics to conclusions according to your causal diagram, ideally in the results or intro not at the end in the limitations.

The diagrams and texts referred in the previous comment are relevant to this comment. In addition, we have expanded the discussion of deviations from direct assortment (which also indicates deviations from purely causal associations) on line 324-346:

Although the phenotypic model could be falsified, the underlying mechanisms remain elusive. Both indirect assortment and social stratification³⁸ could increase in-law correlations disproportionately and explain our observations. In any case, partner resemblance is not solely due to assortment based on the observed phenotypes. Whether parts of the partner correlations in mental health are due to causal influences on partner choice remains to be determined. Identifying the traits that actively determine assortment is an important question for future studies. It might be more strongly related to general vulnerability to psychopathology³¹ than to specific disorders. Due to the strong cross-trait assortment, such causal effects may be more plausible at the level of general mental health, rather than for specific diagnoses. A previous study indicated that partner similarity in many traits was driven by assortment on a few key traits³⁸, but it did not include mental disorders. Future studies may explore whether partner resemblance across many traits can be more parsimoniously explained by assortment on one or a small number of dimensions.

Indirect assortment need not be based on symmetric assortment on a manifest phenotype. Measurement error can be indistinguishable from indirect assortment on an unknown trait. Assortment may then be said to be direct for the true values of a trait, but indirect for an imperfect indicator. As measurement error is widespread and relatively easy to estimate, accounting for measurement error could improve future studies on assortment. Indirect assortment could also be

related to impression management, whereby partner selection could take place on successful misrepresentations of one's characteristics. This should, however, not influence sibling correlations. Finally, correlations in trait preferences among siblings can increase correlations between distant affines, such as co-siblings-in-law³². Hence, models of preferences may be needed to fully understand similarities in wider family networks.

Regarding measurement, we have also extended the discussion of the use of health records in the limitations section of the discussion, line 394-404:

First, the medical records are proxies for actual health conditions, as not all individuals with health issues seek medical care. This prevented the study of conditions below the threshold of medical attention. This issue is reduced as the tetrachoric correlations model these thresholds. Also, our use of primary care data captures a larger proportion of cases than specialist care data alone³⁹, which has been used in previous studies². This further mitigates potential biases. We could only study somatic conditions that were common among parents-to-be in young adulthood. The results are not necessarily representative for other somatic health conditions; in particular, assortment on rare health conditions is unknown. This also prevented the study of health conditions with a higher average age of onset, such as cardiometabolic conditions and cancers. However, conditions that develop after couple formation cannot directly influence its composition.

3. A clear description of the terminology used in the discussion, if you say “partner correlations in MH after accounting for EA/GPA” what do you mean do you mean you conclude either trait specific homogamy and/or assortment? If you say “partners selection” what do you mean? Only assortment but not homogamy? Selection implies choice and or individual action to the reader.

We refer to partner correlations and residual partner correlations when the causal process is unknown or not the topic. With “partner selection”, we refer to similarity at the time of relationship initiation, that is, direct and indirect assortment and social stratification. We have taken several steps to clarify our terminology:

- We assume the comment on “accounting for” refers to the abstract, line 29-30, which has been rewritten as.

Adjusting for GPA and EA reduced partner correlations in health with 30-40%.

- Figure 1 clearly defines our terminology. Whereas selection could imply active choice, partners can be selected within social strata, that is, we use the term to include all similarities at the time the couple was formed.
- In the introduction, we state that convergence is not a form of assortment, clarifying that the other processes are forms of assortment, line 94-95:

Convergence is not a form of assortment, but an alternative explanation of partner similarity.

This paper is far more than simple cross section analysis, you have done outstanding work trying to learn about complex mechanisms with serious consequences for both biomedical research (psychiatric genetics) and the social sciences, I hope you can be equally rigorous about your causal model, your assumptions therein, and their limitations, if asked to review again I wont be satisfied by a simple summation of these limitations I think its vital the reader understands the causal model you imply, what you can and cannot test in those models given the data, and at what points you need to make some leaps of fate 9i.e. assumptions).

We thank the reviewer for the enthusiastic overall evaluation and for helpful comments. We believe the changes outlined above clarify our causal reasoning.

Minor:

Line 175: “contract” is supposed to be contrast, I think?

This has been corrected.

Reviewer #2 (Remarks to the Author):

I congratulate the authors for the very thorough revision. The paper has strengthened a lot and I'm happy how my comments were addressed.

One minor point is that I mistakenly mentioned hypergeometric test, but I meant binomial test to see whether the number of nominally significant deviations from 1 are significant. This is the following logic: if one observes, say, 15 out of 22 tests with P-value <0.05, the probability to observe 15 or more such tests can be computed as `pbinom(15,22,.05,lower.tail = FALSE)` [R command]. This gives an idea of overall significance of the findings. But given that the authors have applied FDR correction, it is also fine.

I'm not sure what test the authors used to compare the medians of two sets of correlations. ["The correlations between partners' mental health conditions increased notably from a median of 0.14 in prospective analyses to a median correlation of 0.25 in cross-sectional analyses (Δ -2LL = 211.40, Δ df=10, $p < 1.00e-99$)."] It looks like likelihood ratio test, but which are the two models (then it should be " $-2*\Delta LL$ ")?

We thank the reviewer for their insightful and useful comments. We apologize for any confusion caused by our description of the likelihood test. In the manuscript, we are not testing whether the medians of the two sets of correlations are the same. Instead, we test whether two paired sets of correlations can be constrained to be equal. We compare a model with 20 freely estimated correlations (two correlations for each of 10 disorders) to a model with 10 freely estimated correlations (for each of the 10 disorders, the cross-sectional and prospective correlation is the same). The likelihood ratio test comparing these models yields a chi-square statistic with 10 degrees of freedom, corresponding to the 10 constraints imposed. We have updated the manuscript to clarify this point. We also changed Δ -2LL to $-2\Delta LL$ in the referred text and a similar occurrence in the following sentence.

Please see line 165-173:

The correlations between partners' mental health conditions increased notably from prospective analyses, where the median correlation was 0.14, to cross-sectional analyses, where the median correlation was 0.25. We tested whether the cross-sectional and prospective correlations differed by comparing to models: one where the two correlations were estimated freely for each health condition, and another model where they were constrained to be equal. The prospective and cross-sectional correlations could not be to be equal for the 10 mental health conditions ($-2\Delta LL = 211.40$, Δ df=10, $p < 1.00e-99$). For somatic health conditions, the increases were more modest, with a median increase from 0.04 to 0.06, but the correlations could not be set to be equal ($-2\Delta LL = 63.05$, Δ df = 10, $p = 9.55e-10$).

Reviewer #3 (Remarks to the Author):

I am satisfied with the authors' revisions. The manuscript is much stronger and will make a valuable contribution to the current literature.

We thank the reviewer for evaluating our work and for the comments they have previously provided.

Reviewer #5 (Remarks to the Author):

1. The authors have now included a roadmap, presented in Figures 1 and 7, which effectively outlines all the potential mechanisms of partner assortment. This addition is very helpful. We have a few minor suggestions regarding these figures and the accompanying text: 1) In text of Fig 1E add 'N' as follows: "Furthermore, partners could share environments and experiences (N)"; 2) The authors may consider changing the order of the Figures A-D in Fig 7 to be consistent with the order of the those in Fig 1, i.e. Fig 7A: direct assortment, Fig 7B: indirect assortment, Fig 7C: social stratification, Fig 7D: convergence; 3) Should 'E' in Fig 7A actually be 'U' as in Fig 1B? Please also define in Fig 7 legend.; 4) The description of Fig 7 (line 522-556) mentions 'rs' a few times wrt Fig 7C and 7D, but this is called 'rm' in the Figure rather than 'rs' , please correct and make consistent.

We thank the reviewer for very thorough reading of our manuscript and for spotting these mistakes and inconsistencies. 1) This is now implemented. 2) We have re-labelled Figure 7 and re-ordered the text in the methods accordingly. This means swapping B and C, while A and D remains in the original place. 3) We agree that this should be consistent, but as there is a strong tradition for using E as the residual, we instead changed U to E in Figure 1B. As the residuals are different in Figure 1B and 1C, we added "The residuals E can be correlated if assortment on B does not fully explain partner similarity in A" to the explanation of Figure 1C. We also updated the figure legend of Figure 7. "Panel C illustrates indirect assortment, where partner similarities in A is due to assortment on the latent variable M and siblings may also correlate in residual variance (E)." 4) We have changed three occurrences of r_s (observed sibling correlation) to r_m (sibling correlation in the latent factor M) in the methods section.

2. In terms of the potential impact of convergence, the cross-sectional partner correlations (2015–2019) are higher than the prospective unadjusted partner correlations. As a result of convergence, participants and partners are becoming more similar over time, which increases the partner correlation and reduces the in-law inflation factor (IIF). However, when compared with the IIF values in Supplementary Table 3, 6 out of the 22 IIFs for the cross-sectional measurements are higher than those for the prospective measurements. How do the authors explain this? Furthermore, if data on the duration of partner cohabitation is available the authors may consider adjusting for the effects of convergence?

The reviewer is right that all else equal, convergence is expected to lead to reduced IIF in realistic scenarios, as detailed in the methods. Indeed, for 14 of the 20 conditions, IIF was reduced between the analyses, but increased for 6 health conditions. In short, we believe this could be due to other co-occurring processes, such as developments in sibling similarity, and statistical fluctuations. The IIF alone cannot decide which process underlines deviations from phenotypic assortment, and this is the reason we do not directly compare these to determine convergence – only deviations from phenotypic assortment. The p-values refer to deviations from phenotypic assortment, whereas we do not explicitly test differences in IIF in the two analyses. If they are different, it cannot necessarily be attributed to convergence.

For three of the six conditions, IIF is not statistically significantly different from 1.00 in either of the tests, and we would not put much emphasis on the nominal difference (e.g. 0.97 vs 1.03 for hyperkinetic disorder). Furthermore, for the somatic condition, which are 4 of 6 conditions with increases in IIF, the partner correlation is almost the same in the two analyses, increasing with at most 0.02. Without increasing partner correlations, we do not expect IIF to be reduced, and fluctuations in sibling and in-law correlations is likely what lead to somewhat difference IIF estimates. For phobias, the partner correlation goes from 0.09 to 0.14. However, the prospective in-law correlation is 0.03 [-0.02, 0.08] and the cross-sectional 0.07 [0.03, 0.11]. This doubling of estimate affects IIF, but the confidence intervals of the in-law correlations are largely overlapping.

We agree that data on partner cohabitation would be useful for assessing effects of convergence; unfortunately, such information is not available in our datafile.

Rujia Wang & Harold Snieder

Thank you for your valuable contribution to our manuscript.

Reviewer #6 (Remarks to the Author):

We thank this reviewer and wish them luck with their training.